# A Theoretical Framework for Grokking: Interpolation followed by Riemannian Norm Minimisation

**Etienne Boursier**
INRIA
LMO, Université Paris-Saclay
Orsay, France
`etienne.boursier@inria.fr`

**Scott Pesme**
INRIA
Grenoble, France
`scott.pesme@inria.fr`

**Radu-Alexandru Dragomir**
Télécom Paris
Institut Polytechnique de Paris
Palaiseau, France
`dragomir@telecom-paris.fr`

## Abstract

We study the dynamics of gradient flow with small weight decay on general training losses $F : \mathbb{R}^d \to \mathbb{R}$. Under mild regularity assumptions and assuming convergence of the unregularised gradient flow, we show that the trajectory with weight decay $\lambda$ exhibits a two-phase behaviour as $\lambda \to 0$. During the initial fast phase, the trajectory follows the unregularised gradient flow and converges to a manifold of critical points of $F$. Then, at time of order $1/\lambda$, the trajectory enters a slow drift phase and follows a Riemannian gradient flow minimising the $\ell_2$-norm of the parameters. This purely optimisation-based phenomenon offers a natural explanation for the *grokking* effect observed in deep learning, where the training loss rapidly reaches zero while the test loss plateaus for an extended period before suddenly improving. We argue that this generalisation jump can be attributed to the slow norm reduction induced by weight decay, as explained by our analysis. We validate this mechanism empirically on several synthetic regression tasks.

## 1 Introduction

Strikingly simple algorithms such as gradient methods are a driving force behind the success of deep learning. Nonetheless, their remarkable performance remains mysterious, and a full theoretical understanding is lacking. In particular: (i) convergence to low training loss solutions on non-convex objectives is far from trivial, and (ii) it is unclear why the resulting solutions generalise well [Zhang et al., 2017]. These questions are accompanied by a range of surprising phenomena that arise during training. One such intriguing behaviour is known as the *grokking phenomenon*, which we explore in this work. Coined by Power et al. [2022], this term describes a two-phase pattern in the learning curves: first, the training loss rapidly decreases to zero, while the test loss plateaus at a certain value. This is followed by a second phase, where the training loss remains zero, but the test loss steadily decreases, leading to a final improved generalisation performance, as depicted in Figure 1 (left).

In this paper, we propose a novel theoretical perspective to explain this phenomenon. By examining the gradient flow dynamics **with weight decay**, we show that, in the limit of vanishing weight decay, we can fully describe the trajectory of the model parameters. Specifically, we prove that the training process can be decomposed into two distinct phases. In the first phase, the gradient flow follows the unregularised path, converging to a manifold of critical points of the training loss. In the second, the trajectory enters a slow drift phase, where the weights move along this manifold, driven by weight

39th Conference on Neural Information Processing Systems (NeurIPS 2025).

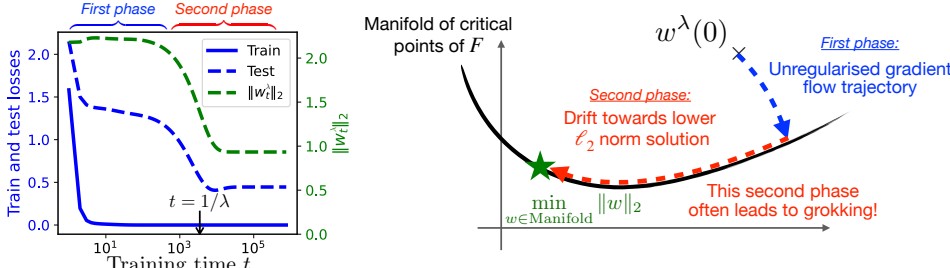

Figure 1: Gradient flow with small weight decay $\lambda$. *(Left)* A typical example of grokking: the training loss rapidly drops to zero, while the test loss plateaus for a long period before eventually decreasing—coinciding with a steady drop in the $\ell_2$-norm of the weights. *(Right)* Schematic illustration in parameter space $\mathbb{R}^d$ of the optimisation behaviour described in Theorem 1. The trajectory $w^\lambda(t)$ initially follows the unregularised gradient flow and converges to a manifold of critical points of $F$ (fast dynamics). At time $t \approx 1/\lambda$, the regularisation term becomes dominant and induces a slow drift along this manifold toward a lower $\ell_2$-norm solution (slow dynamics).

decay, gradually reducing their $\ell_2$ norm, as illustrated in Figure 1 (right). We argue that this slow decrease in the weight norms explains the grokking phenomenon, as smaller weight norms are often correlated with better generalisation. To convey the main intuition, we state below an informal version of our result, describing the full trajectory of the gradient flow with small weight decay.

**Informal statement of the result.** For a generic training loss $F : \mathbb{R}^d \to \mathbb{R}$ satisfying some regularity assumptions, we consider the gradient flow $w^\lambda$ regularised with weight decay: $\dot{w}^\lambda(t) = -\nabla F(w^\lambda(t)) - \lambda w^\lambda(t)$. Under the assumption that the unregularised gradient flow trajectory is bounded, we prove the following.

**Theorem 1** (Main result, informal). *As the weight decay parameter $\lambda$ is taken to $0$, the trajectory $w^\lambda(t)$ can be seen as a composition of two coupled dynamics:*

1. *(**Fast dynamics driven by F given by Proposition 1**) In a first phase, the weights follow the unregularised gradient flow and converge to a manifold of critical points of $F$.*

2. *(**Slow dynamics driven by the weight decay given by Proposition 2**) At time $t \approx 1/\lambda$, the iterates start slowly drifting along this manifold, following a Riemannian gradient flow that decreases the $\ell_2$-norm of the weights.*

**Link with the grokking phenomenon.** Note this is a purely optimisation result: no statistical assumptions are made, and it *a priori* does not imply any improvement in test loss during the slow phase. However, it provides a natural explanation for the grokking phenomenon. Indeed, in practice, for many deep learning models with random initialisation, gradient flow converges to a global minimiser of the training loss. When this solution generalises poorly—as is often the case with large initial weights, in the so-called *lazy* regime [Chizat et al., 2019]—the subsequent slow drift along the critical manifold, driven by weight decay, decreases the $\ell_2$-norm of the solution and simplifies it in the second phase. Since lower weight norms often correlate with better generalisation [Bach, 2017, Liu et al., 2022c, D'Angelo et al., 2024], this offers a convincing explanation for the delayed improvement in test performance. We discuss various settings where this behaviour is observed in Section 5.

## 2  Related work

**Grokking in experimental works.** The term *grokking* was originally coined by Power et al. [2022], which studied a two-layer transformer trained with weight decay on a modular addition task. They observed that the network quickly fits the training data while generalising poorly, followed much later by a sudden transition to near-perfect generalisation. Following this work, many studies have investigated modular addition tasks to better understand the mechanisms underlying this phenomenon [Nanda et al., 2023, Gromov, 2023]. However, grokking has been observed far beyond this setting. For instance, Barak et al. [2022] showed that training a neural network to learn parities exhibits a similar delayed generalisation pattern. In Liu et al. [2022c], grokking was induced across a broad range of tasks, including image classification and sentiment analysis, by using small datasets, large initialisations, and weight decay. Other settings and architectures where grokking-like

behaviour appears include matrix factorisation [Lyu et al., 2023] and learning XOR-clustered data with a ReLU network [Xu et al., 2023]. Finally, it is worth noting that this delayed transition in generalisation was already observed in earlier works, as clearly illustrated in Figure 3 of Chizat and Bach [2020]. More recently, Jeffares and van der Schaar [2025] argued that grokking may not be so central to Deep Learning and may only appear in very specific situations. However, we still believe its *a priori* counter-intuitive aspect is worth investigating and might lead to theoretical understandings that go beyond what is currently referred by grokking.

**Grokking as the transition between lazy and rich regimes.** Several works have framed grokking as the transition between the lazy and rich regimes. The lazy regime, also called the NTK regime, was introduced by Jacot et al. [2018]. It typically arises when the network is trained from large initialisations [Chizat et al., 2019], and corresponds to a setting where zero training loss can be quickly achieved, but often with poor generalisation performance. In contrast, the rich regime (also called the feature learning regime) corresponds to a setting where the network actively learns new internal features during training. In the classification setting, Lyu and Li [2019] show that the rich regime is always attained for homogeneous parameterisations, and similarly, Chizat and Bach [2020] provide an analogous result for infinitely wide two-layer networks. In this context, Lyu et al. [2023] and Kumar et al. [2024], followed by Mohamadi et al. [2024], offer a theoretical perspective on grokking as the transition from the lazy regime to the rich regime during training: initially, the predictor quickly converges towards the NTK solution, and later escapes this regime to reach a better generalising solution, driven by the effects of implicit regularisation and/or weight decay.

**The role of weight decay.** The role of weight decay in grokking remains somewhat debated. While many of the original works exhibiting the phenomenon include weight decay [Power et al., 2022, Liu et al., 2022b], grokking has also been observed without [Chizat and Bach, 2020, Xu et al., 2023], as strongly emphasised by Kumar et al. [2024]. However, as shown in Lyu et al. [2023], the transition tends to occur much later and to be less sharp without weight decay. In this context, weight decay can be interpreted as a factor that triggers or accelerates the transition from the lazy to the rich regime. While grokking can be observed in classification tasks even without weight decay—thanks to the algorithm's implicit bias—to the best of our knowledge, it cannot occur in regression tasks unless weight decay is used. Of particular relevance to our work, Liu et al. [2022c] propose an intuitive explanation of grokking that is based on weight decay: during the first phase, the model rapidly converges to a poor global minimum; during the second, slower phase, weight decay gradually steers the iterates toward a lower-norm solution with better generalisation properties. While appealing, this explanation remains informal and lacks rigorous theoretical support. In this work, we provide a formal analysis of the optimisation dynamics underlying the grokking phenomenon: an initial fast phase leads to convergence toward the solution associated with the lazy regime, followed by a slower second phase that drives convergence toward the solution characteristic of the rich regime.

**Drift on the interpolation manifold.** Many theoretical works in the machine learning community have studied the training dynamics of gradient methods in overparameterised neural networks, where the set of zero training loss solutions forms a high-dimensional manifold. In this context, leveraging results from dynamical systems theory [Katzenberger, 1990], the work of Li et al. [2021] describes the drift dynamics induced by stochastic noise after stochastic gradient descent (SGD) reaches the manifold. This analysis was further extended by Shalova et al. [2024]. Leveraging a similar stochastic differential framework, Pillaud-Vivien et al. [2022] precisely characterises this drift in the setting of diagonal linear networks, and proves that it leads to desirable sparsity guarantees. Much in the spirit of our work, although outside the deep learning context, Fatkullin et al. [2010] derive stochastic differential equations that describe the dynamics of systems with small random perturbations on energy landscapes with manifolds of minima, illustrating how the system first converges to the manifold and then drifts along it.

Our analysis builds upon this framework popularized by Li et al. [2021] and tracing back to Katzenberger [1990]. Our contribution differs from previous work in both focus and scope. Whereas previous analyses attribute the second, slower learning phase to stochastic effects, we identify a deterministic mechanism–namely, a slow drift induced by regularization–and link it directly to the grokking phenomenon, a connection that, to our knowledge, has not been previously explored. From a technical perspective, our setting is simpler yet enables a more detailed analysis. Rather than appealing to results from Katzenberger [1990] as a black box, we provide a simpler, self-contained proof building on Falconer [1983]. Furthermore, unlike prior analyses that assume initialization near the interpolation manifold, our framework accommodates arbitrary initialisation. We rigorously

characterise the initial convergence toward the manifold and the subsequent transition to the drift phase along it, which constitutes the main technical novelty of our analysis.

# 3 Setting and preliminaries

We consider a loss function $F : \mathbb{R}^d \to \mathbb{R}_+$ which is sufficiently smooth, as stated in Assumption 1 below. Typical examples include the least square loss over some training dataset, where the parameters to optimise represent the weights of some neural network architecture. For a given $\lambda > 0$, we define the regularised loss $F_\lambda$ as:

$$F_\lambda(w) := F(w) + \frac{\lambda}{2}\|w\|_2^2, \qquad \forall w \in \mathbb{R}^d.$$

Initialising the parameters from $w_0 \in \mathbb{R}^d$ (independently of $\lambda$), we then consider the gradient flow $w^\lambda$ over the regularised loss for any $\lambda > 0$, as the solution of the differential equation

$$\dot{w}^\lambda(t) = -\nabla F_\lambda(w^\lambda(t)) \quad \text{and} \quad w^\lambda(0) = w_0. \tag{1}$$

Gradient flow is the limit dynamics of (stochastic) gradient descent as the learning rate goes to 0. For $\lambda = 0$, we denote $w^{\mathrm{GF}}$ the gradient flow on the unregularised loss:

$$\dot{w}^{\mathrm{GF}}(t) = -\nabla F(w^{\mathrm{GF}}(t)) \quad \text{and} \quad w^{\mathrm{GF}}(0) = w_0. \tag{2}$$

In the remaining of the paper, we consider the following assumption on the objective function.

**Assumption 1.** *The function $F$ is $\mathcal{C}^3$ on $\mathbb{R}^d$, its third derivative is locally Lipschitz and $F$ is definable in an o-minimal structure. Moreover, the solution $w^{\mathrm{GF}}$ to the gradient flow ODE (2) is bounded.*

The regularity conditions on $F$ ensure that, for any $\lambda \geq 0$, the gradient flow ODE has a unique solution, which is defined for all $t \geq 0$. This is a consequence of the Picard–Lindelöf theorem and boundedness of the trajectories. Definability in the o-minimal sense guarantees that bounded gradient flow trajectories converge to a limit point [Kurdyka, 1998, Thm. 2]. This is a mild assumption satisfied by most functions arising in applications, such as polynomials, logarithms, exponentials, subanalytic functions, and finite combinations of those; see Coste [1999], Bolte et al. [2007] for more details. Overall, Assumption 1 is pretty mild and holds for all architectures (e.g., neural networks or transformers) that use differentiable activations.

Finally, note that the boundedness assumption on the unregularised flow excludes classification settings where the network can perfectly separate the data, since in such cases the unregularised iterates diverge.

## 3.1 Stationary manifold and Riemannian flow

Assumption 1 guarantees that the gradient flow converges to a limit point $w^{\mathrm{GF}}_\infty := \lim_{t\to\infty} w^{\mathrm{GF}}(t)$, which is a stationary point of $F$. In typical scenarios of training overparameterised models, stationary points are not isolated, but form continuous sets [Cooper, 2021].

**Definition** (Definition of the manifold $\mathcal{M}$). *We define $\mathcal{M}$ to be the largest connected component of $\nabla F^{-1}(0)$ containing $w^{\mathrm{GF}}_\infty$, where $\nabla F^{-1}(0)$ corresponds to the set of stationary points of $F$.*

Our key assumption is that $\mathcal{M}$ forms a smooth manifold, and that it contains only local minimizers (and not saddle points). Additionnally, we impose that the non-zero eigenvalues of the Hessian on $\mathcal{M}$ are lower bounded by some constant $\eta > 0$. Following the terminology of Rebjock and Boumal [2024], this is known as the *Morse-Bott property*.

**Assumption 2.** *$\mathcal{M}$ is a smooth submanifold of $\mathbb{R}^d$ of dimension $k \in [d]$, i.e. for any $w \in \mathcal{M}$, $\mathrm{rank}(\nabla^2 F(w)) = d - k$. Also, there exists $\eta > 0$ such that for any $w \in \mathcal{M}$, all non-zero eigenvalues of $\nabla^2 F(w)$ are lower bounded by $\eta$.*

As stated in Rebjock and Boumal [2024], for $\mathcal{C}^2$ functions this property is equivalent to the *Kurdyka–Łojasiewicz*, also known as *Polyak–Łojasiewicz* (PL) condition locally around $\mathcal{M}$ [Kurdyka, 1998, Bolte et al., 2009]. It implies that (i) every point in $\mathcal{M}$ is a local minimiser, and (ii) all gradient flow trajectories are locally attracted towards $\mathcal{M}$. This stability property is essential for proving regularity of the flow map $\Phi$ in Section 4. Let us discuss the relevance of Assumption 2 in the context of overparameterised machine learning.

- **Convergence to a local minimiser:** our assumption rules out the possibility that $w^{\mathrm{GF}}$ converges to a saddle point of $F$. This is justified by a large number of works showing that gradient methods avoid saddle points for *almost all* initialisations. In particular, Lee et al. [2016, 2019] prove it under the assumption that all saddle points of $F$ are strict. Although their result only holds for discrete-time gradient descent, the underlying argument can be extended to gradient flow (the proof relies on the Stable Manifold Theorem for dynamical systems, which also holds in continuous time [Teschl, 2012, §7]).
- **Morse-Bott/Łojasiewicz property:** this is a common assumption in the analysis of gradient flow dynamics for overparameterised networks [Li et al., 2021, Fatkullin et al., 2010, Shalova et al., 2024]. Note that for general models, the critical set may not form a manifold everywhere. However, it is often possible to show that the manifold structure holds locally, i.e., on *most* of the space, excluding some degenerate points[1]; see Section 5 for examples and Liu et al. [2022a] for a generic result. Moreover, the results derived from such an assumption are generally very representative of empirical observations, as can be seen in Section 5.

**Riemannian gradient flow on $\mathcal{M}$.** We endow $\mathcal{M}$ with the standard Euclidean metric. For any differentiable function $h : \mathbb{R}^d \to \mathbb{R}$, we denote by $\mathrm{grad}_{\mathcal{M}} h$ the Riemannian gradient on $\mathcal{M}$ of the function $h$ defined as follows:

$$\mathrm{grad}_{\mathcal{M}} h : \begin{array}{l} \mathcal{M} \to \mathbb{R}^d \\ w \mapsto P_{T_{\mathcal{M}}(w)}(\nabla h(w)), \end{array}$$

where $P_{T_{\mathcal{M}}(w)}$ is the orthogonal projection on the tangent space to $\mathcal{M}$ at $w$. Under Assumption 2, typical properties of smooth manifolds imply that $T_{\mathcal{M}}(w) = \mathrm{Ker}(\nabla^2 F(w))$ [see e.g., Boumal, 2023, for a detailed introduction to optimisation on manifolds]. Using this notion of Riemannian gradient, we study the Riemannian gradient flow for some objective function $h$ and initialization $w_{\mathcal{M}} \in \mathcal{M}$, defined as the curve $w$ satisfying,

$$\dot{w}(t) = -\mathrm{grad}_{\mathcal{M}} h(w(t)) \quad \text{and} \quad w(0) = w_{\mathcal{M}}. \tag{3}$$

By construction of the Riemannian gradient, the trajectory of any solution of this ODE necessarily belongs to $\nabla F^{-1}(0)$, and therefore to $\mathcal{M}$, since $\mathcal{M}$ is a maximal connected component. If $h$ is $\mathcal{C}^2$ and has compact sublevel sets, Assumptions 1 and 2 guarantee that there exists a unique solution to Equation (3) and that it is defined on $\mathbb{R}_+$.

## 4 Grokking as two-timescale dynamics

This section states our main results, where we characterise the two-timescale dynamics of the regularised gradient flow (1) in the limit $\lambda \to 0$. In Section 4.1 we describe the first phase, the *fast dynamics*, where $w^\lambda$ approximates the unregularised gradient flow solution on finite time horizons. Section 4.2 then identifies the second, *slow dynamics* happening at arbitrarily large time horizons, where $w^\lambda$ follows the Riemannian flow of the $\ell_2$ norm on the manifold $\mathcal{M}$ of stationary points.

### 4.1 Fast dynamics

A first simple observation is that as $\lambda \to 0$, $F_\lambda \to F$ uniformly on any compact of $\mathbb{R}^d$. From there, it seems natural that $w^\lambda$ should converge, at least pointwise, to $w^{\mathrm{GF}}$ as $\lambda \to 0$. A Grönwall argument indeed allows to characterise the first, fast timescale dynamics given by Proposition 1 below.

**Proposition 1.** *If Assumption 1 holds, then for all $T \geq 0$, $w^\lambda \underset{\lambda \to 0}{\longrightarrow} w^{GF}$ in $(\mathcal{C}^0([0,T], \mathbb{R}^d), \|\cdot\|_\infty)$*

Importantly, uniform convergence only holds on finite time intervals of the form $[0, T]$, but is not true on $\mathbb{R}_+$. More precisely, grokking is observed when the two limits cannot be exchanged: $\lim_{\lambda \to 0^+} \lim_{t \to \infty} w^\lambda(t) \neq \lim_{t \to \infty} \lim_{\lambda \to 0^+} w^\lambda(t) = \lim_{t \to \infty} w^{\mathrm{GF}}(t)$. In Figure 1, the endpoint of the red arrow corresponds to this first limit, while the endpoint of the blue arrow corresponds to the second. This distinction highlights a key aspect of grokking: the dynamics evolve on two different timescales. Initially, as described by Proposition 1, the regularised flow tracks the unregularised one. But at much larger time horizons, the regularised dynamics begin to diverge.

---

[1]While our results are stated for such a global Morse-Bott assumption, they could be directly extended to local Morse-Bott assumptions, provided that the iterates remain in the non-degenerate region.

## 4.2 Slow dynamics

The second part of the dynamics is harder to capture, since it happens at a time approaching infinity, when $\lambda$ approaches zero. It is done using theory of singularly perturbed systems. In the following, we associate to the unregularised flow function a mapping $\phi : \mathbb{R}^d \times \mathbb{R}_+ \to \mathbb{R}^d$ satisfying

$$\phi(w,t) = w - \int_0^t \nabla F(\phi(w,s)) \, \mathrm{d}s, \qquad \forall (w,t) \in \mathbb{R}^d \times \mathbb{R}_+.$$

Note that $\phi(w,t)$ simply corresponds to the solution of the gradient flow of Equation (2) at time $t$ when initialised in $w$. When possible to define—i.e., when the gradient flow admits a limit point in $\mathbb{R}^d$—we define the mapping $\Phi$ as

$$\Phi(w) = \lim_{t \to \infty} \phi(w,t). \tag{4}$$

Thanks to Assumption 1, the unregularised flow initialised in $w_0$ admits the limit point $\Phi(w_0)$, which is necessarily a stationary point of the loss, i.e., $\nabla F(\Phi(w_0)) = 0$. Assumption 2 then ensure that the mapping $\Phi$ is defined and $\mathcal{C}^2$ on some neighbourhood of $\mathcal{M}$, thanks to a result of Falconer [1983].

**Lemma 1.** *If Assumptions 1 and 2 hold, there exists an open neighbourhood $U$ of $\mathcal{M}$ such that $\Phi$ is defined and $\mathcal{C}^2$ on $U$.*

Now the mapping $\Phi$ is well defined on a neighbourhood of $\mathcal{M}$, it can be used to describe the limit of the slow dynamics. For $\lambda > 0$, we let $\tilde{w}^\lambda : t \mapsto w^\lambda(t/\lambda)$. We indeed have to adequately "speed-up" time to capture its behaviour. Notably, $\tilde{w}^\lambda$ satisfies the following differential equation:

$$\dot{\tilde{w}}^\lambda(t) = -\tilde{w}^\lambda(t) - \frac{1}{\lambda} \nabla F(\tilde{w}^\lambda(t)) \quad \text{and} \quad \tilde{w}^\lambda(0) = w_0. \tag{5}$$

Our goal in this section is to study the limit function $\lim_{\lambda \to 0} \tilde{w}^\lambda$. Intuitively, the $\frac{1}{\lambda} \nabla F$ term in Equation (5) will enforce this limit to stay on the stationary manifold $\mathcal{M}$ for any $t > 0$. The slow dynamics will be shown to approximate the Riemannian flow of the squared Euclidean norm on the stationary manifold $\mathcal{M}$ for $t > 0$. This limit flow is defined by $\tilde{w}^\circ$, which is the solution of the following differential equation on $\mathbb{R}_+$, for the function $\ell_2 : w \mapsto \|w\|_2^2/2$,

$$\dot{\tilde{w}}^\circ(t) = -\mathrm{grad}_{\mathcal{M}} \, \ell_2(\tilde{w}^\circ(t)) \quad \text{and} \quad \tilde{w}^\circ(0) = \Phi(w_0). \tag{6}$$

Recall that the Riemannian gradient is $\mathrm{grad}_{\mathcal{M}} \, \ell_2(w) = P_{\mathrm{Ker}(\nabla^2 F(w))}(w)$ for any $w \in \mathcal{M}$. Denoting $D\Phi_w$ the differential of $\Phi$ at $w$, Li et al. [2021, Lemma 4.3] proved that for any $w \in \mathcal{M}$, $P_{\mathrm{Ker}(\nabla^2 F(w))} = D\Phi_w$, i.e., the differential of $\Phi$ at $w$ is given by the orthogonal projection onto the kernel space of the Hessian of $F$. In consequence, $\tilde{w}^\circ$ also satisfies the following differential equation:

$$\dot{\tilde{w}}^\circ(t) = -D\Phi_{\tilde{w}^\circ(t)}(\tilde{w}^\circ(t)) \quad \text{and} \quad \tilde{w}^\circ(0) = \Phi(w_0).$$

Using this alternative description of $\tilde{w}^\circ$, we can now prove our main result, given by Proposition 2.

**Proposition 2.** *If Assumptions 1 and 2 hold, then for all $T, \varepsilon > 0$, we have $\tilde{w}^\lambda \xrightarrow[\lambda \to 0]{} \tilde{w}^\circ$ in $(\mathcal{C}^0([\varepsilon, T], \mathbb{R}^d), \|\cdot\|_\infty)$, where $\tilde{w}^\circ$ is the unique solution on $\mathbb{R}_+$ of the differential equation (6).*

Proposition 2 states that the slow dynamics $\tilde{w}^\lambda$ converges uniformly to $\tilde{w}^\circ$ as $\lambda \to 0$ on any compact interval of the form $[\varepsilon, T]$. Note that excluding 0 from this interval (i.e., $\varepsilon > 0$) is necessary. Indeed, uniform convergence cannot happen on an interval of the form $(0, T]$, since $\tilde{w}^\lambda(0) = w_0$ for any $\lambda > 0$ and $\tilde{w}^\circ(0) = \Phi(w_0)$. In particular, Proposition 2 leads to pointwise convergence of $\tilde{w}^\lambda$: we have

$$\begin{cases} \lim_{\lambda \to 0} \tilde{w}^\lambda(0) = w_0, \\ \lim_{\lambda \to 0} \tilde{w}^\lambda(t) = \tilde{w}^\circ(t) & \text{if } t > 0. \end{cases}$$

This limit function $\lim_{\lambda \to 0} \tilde{w}^\lambda$ is non-continuous at 0. Indeed the whole *fast dynamics*, which follows the unregularised flow, happens at that 0 point in the limit $\lambda \to 0$. On the other hand, Proposition 2 describes the second phase of the dynamics, starting from the convergence point of the unregularized flow $\Phi(w_0)$ – at the rescaled time $0^+$ – and following the Riemannian flow on $\mathcal{M}$.

Note that, once the junction between the slow and fast dynamics is carefully handled via Lemma 2 in Appendix A.2, Proposition 2 can be derived from Fatkullin et al. [2010, Theorem 2.2], which heavily relies on the technical result of Katzenberger [1990]. However, for the sake of completeness and readability, we provide a concise and self-contained proof of Proposition 2, avoiding the use of heavyweight methods from Katzenberger [1990] and relying on weaker assumptions.

**Sketch of proof.** We here provide a sketch of proof with the key arguments leading to Proposition 2. Its complete and detailed proof can be found in Appendix A.3. We first define the shifted slow dynamics $\tilde{v}^\lambda$ for any $t \geq 0$ as $\tilde{v}^\lambda(t) = \tilde{w}^\lambda(t + t(\lambda))$, where $t(\lambda)$ is the "junction point" between the two dynamics given by Lemma 2 in Appendix A.2, and satisfies $\lim_{\lambda \to 0} t(\lambda) = 0$. Using Lemma 2, $\tilde{v}^\lambda$ then follows the same differential equation as $\tilde{w}^\lambda$, with an initial condition now satisfying $\lim_{\lambda \to 0} \tilde{v}^\lambda(0) = \Phi(w_0) \in \mathcal{M}$. While the dynamics of $\tilde{v}^\lambda$ might be hard to control as $\lambda \to 0$, it is easier to control the one of $\Phi(\tilde{v}^\lambda)$. Using the chain rule, we indeed have

$$\dot{\Phi}(\tilde{v}^\lambda(t)) = -D\Phi_{\tilde{v}^\lambda(t)} \cdot \left( \tilde{v}^\lambda(t) + \frac{1}{\lambda}\nabla F(\tilde{v}^\lambda(t)) \right).$$

Then using the fact that for any $w$ in a neighbourhood of $\mathcal{M}$, $D\Phi(w) \cdot \nabla F(w) = 0$ [Li et al., 2021, Lemma C.2], this directly rewrites as

$$\dot{\Phi}(\tilde{v}^\lambda(t)) = -D\Phi_{\tilde{v}^\lambda(t)} \cdot \tilde{v}^\lambda(t).$$

Now note that this resembles the differential equation satisfied by $\tilde{w}^\circ$. The two differences being that (i) the initialisation points differ, but $\lim_{\lambda \to 0} \tilde{v}^\lambda(0) = \tilde{w}^\circ(0)$; (ii) the time derivative is on $\Phi(\tilde{v}^\lambda)$ rather than $\tilde{v}^\lambda$ directly. To handle the second point, $\tilde{v}^\lambda$ and $\Phi(\tilde{v}^\lambda)$ obviously converge to the same initialisation point as $\lambda \to 0$. One can then use stability of the manifold $\mathcal{M}$, thanks to Assumption 2, to show that $\sup_{t \in [0,T]} \|\tilde{v}^\lambda(t) - \Phi(\tilde{v}^\lambda(t))\|$ converges to 0 as $\lambda \to 0$. This then allows to conclude. $\square$

**Characterizing the limit of the Riemannian flow.** By monotonicity of its norm, $\tilde{w}^\circ$ is obviously bounded over time. Typical optimisation results then guarantee that the limit set of $\tilde{w}^\circ(t)$ as $t \to \infty$ is contained in the set of critical points of the squared Euclidean norm on the manifold $\mathcal{M}$, given by the KKT points of the following constrained problem:

$$\min_{w \in \mathcal{M}} \|w\|_2^2. \tag{7}$$

The notion of KKT points indeed extend to smooth manifolds [Bergmann and Herzog, 2019], so that under Assumption 2, the KKT points of Equation (7) are given by the points $w^\star \in \mathcal{M}$ satisfying $\mathrm{grad}_{\mathcal{M}}\ell_2(w^\star) = 0$, where we recall $\mathrm{grad}_{\mathcal{M}}\ell_2$ is the Riemannian gradient.

We are then able to show that $w^\lambda$ converges towards the set of KKT points. Note that this does not follow from Proposition 2 alone, as we also need to show that trajectory of $w^\lambda$ remains bounded independently of $\lambda$: we can prove this is true in our case.

**Proposition 3.** *If Assumptions 1 and 2 hold, then for any sequence $(\lambda_k)_{k \in \mathbb{N}}$ such that $\lambda_k \xrightarrow[k \to \infty]{} 0$, the limit points of $(\lim_{t \to \infty} w^{\lambda_k}(t))_{k \in \mathbb{N}}$ are included in the KKT points of Equation (7).*

While Proposition 3 guarantees that $w^\lambda$ gets arbitrarily close to KKT points of Equation (7) as $\lambda$ goes to 0, it does not imply that it has the same limit as $\tilde{w}^\circ$. It is however guaranteed with the additional assumption that $\tilde{w}^\circ(t)$ converges to a strict local minimum of the Euclidean norm on the manifold $\mathcal{M}$.

**Proposition 4.** *Let Assumptions 1 and 2 hold and, assume additionally that $w^\circ(t)$ converges towards a strict local minimum $w^\star$ of the constrained problem (7). Then $\lim_{\lambda \to 0} \lim_{t \to \infty} w^\lambda(t) = w^\star$.*

When the slow limit dynamics on $\mathcal{M}$ converges towards a strict local minimum, Proposition 4 guarantees that, for small enough $\lambda$, $w^\lambda$ gets trapped in the vicinity of this local minimum as $t \to \infty$, allowing us to get a perfect characterisation of $\lim_{\lambda \to 0} \lim_{t \to \infty} w^\lambda(t)$. In particular, this double limit corresponds to the limit of the slow dynamics $\tilde{w}^\circ$, while the permuted limit $(\lim_{t \to \infty} \lim_{\lambda \to 0} w^\lambda(t))$ corresponds to the limit of the fast dynamics $w^{\mathrm{GF}}$, thanks to Proposition 1.

**Role of initialisation scale.** The initialisation scale strongly influences the behavior of the unregularised flow and, consequently, the first phase of training under weight decay. This dependence on scale is well-documented in the literature [Chizat et al., 2019]. While a complete theoretical understanding remains open, it is widely accepted that small initialisation scales correspond to the rich regime, in which implicit bias drives the model toward interpolating solutions with smaller weight norms, typically associated with better generalization. In contrast, large initialisation scales give rise to the lazy or NTK regime, where features change little during training. This regime behaves similarly to random feature models and tends to produce interpolators with weaker generalisation performance.

In our framework, this distinction has a direct consequence. With a small initialisation scale, the point $\Phi(w^\lambda(0))$ reached after the first phase already exhibits a small norm–possibly corresponding

to a KKT point of Equation (7)–so no second grokking phase occurs, as the system has effectively converged. Conversely, with a large initialisation scale, $\Phi(w^\lambda(0))$ retains a large norm, which triggers substantial movement during the second, grokking phase.

**The grokking transition is not sudden!** In the literature, grokking is often described as a "sudden drop" in the validation loss following an extended phase of overfitting. We argue here that this drop only **seems sudden when training time is plotted on a logarithmic scale**, whereas in fact it unfolds over a characteristic duration of order $1/\lambda$, following a plateau of comparable duration $1/\lambda$. Indeed, Proposition 2 predicts that the drop takes place within the time interval $[\varepsilon/\lambda, M/\lambda]$, where $\varepsilon$ is a small time independent of $\lambda$ (think of $\varepsilon$ as the time it takes for $\tilde{w}^\circ$ to move very slightly away from $w_\infty^{\mathrm{GF}}$), and $M$ corresponds to the typical time required for $\tilde{w}^\circ(t)$ to approach its limit $\lim_{t\to\infty} \tilde{w}^\circ(t)$. Prior to this interval, the parameters evolve according to the unregularised flow $w^{\mathrm{GF}}(t)$. Let $M'$ denote the typical time it takes for $w^{\mathrm{GF}}(t)$ to reach $w_\infty^{\mathrm{GF}}$; then the plateau extends over the interval $[M', \varepsilon/\lambda]$. As a result, on a logarithmic time scale, the drop occurs within an interval of length $\ln(M/\varepsilon)$, while the preceding plateau spans roughly $\ln(\varepsilon/\lambda)$ on the same scale. This explains why, as $\lambda \to 0$, the drop appears abrupt compared to the plateau in log-scale plots. In contrast, when viewed in linear time, the drop actually extends over a duration comparable to that of the preceding plateau, producing a markedly different visual impression.

**Comparison with Lyu et al. [2023].** The work most closely related to ours is that of Lyu et al. [2023], which provides a theoretical characterization of the grokking phenomenon as a transition from the NTK regime—i.e., the unregularised flow initialised at large scales—to the *rich regime*, which typically converges to KKT points of Equation (7). However, their analysis does not offer a general optimisation-based perspective on the phenomenon and, in particular, does not account for the slow drift phase along the solution manifold, which we identify and characterise. Moreover, their setting is more restrictive: it assumes specific network architectures with homogeneous parameterisation and requires large initialisation scales. In contrast, our results hold outside the NTK regime and apply across a broader class of settings. In addition, their theoretical guarantees rely on taking the initialisation scale to infinity while simultaneously letting the regularisation strength tend to zero, with both rates polynomially coupled. Their analysis establishes that, for some sufficiently large time $\tilde{t}(\lambda)$, the regularised flow $w^\lambda$ approaches KKT points of Equation (7), but it does not provide guarantees about the asymptotic behavior beyond this time. By contrast, Proposition 3 characterises the limit points of the flow $w^\lambda$, offering a stronger and more complete understanding of its long-term dynamics.

## 5 Examples and experiments

**Linear regression.** Let $F(w) = \|Xw - y\|_2^2$ with $X \in \mathbb{R}^{n\times d}$, and assume that $\min F = 0$. The problem is convex and the set of critical points is the affine subspace $\mathcal{M} = \{w \ : \ Xw = y\}$; Assumption 2 is satisfied globally.

It is well known that unregularised gradient flow $w^{\mathrm{GF}}$ converges to $\mathcal{P}_\mathcal{M}(w_0)$, the projection of the initial point on $\mathcal{M}$ [Lemaire, 1996, Gunasekar et al., 2018]. Then, since $\mathcal{M}$ is convex, the Riemannian flow on $\mathcal{M}$ necessarily converges to the minimal $\ell_2$ norm solution $w^\star = X^+ y$, where $X^+$ denotes the pseudo-inverse. Those two points are different (unless $w_0 = 0$), which leads to grokking, as $w^\star$ is expected to have better generalization properties than $\mathcal{P}_\mathcal{M}(w_0)$ [Bartlett et al., 2020]. In this setting, the trajectories of $w^\lambda$ can be computed explicitly to illustrate the two-timescale dynamics; see Appendix C.

**Matrix completion.** This is a prototypical non-convex problem which is amenable to theoretical analysis. The goal is to recover a matrix $M^\star \in \mathbb{R}^{n\times m}$, which is assumed to be low-rank, from a subset of observed entries in $\Omega \subset [n] \times [m]$, by solving

$$\min_{U\in\mathbb{R}^{n\times r}, V\in\mathbb{R}^{m\times r}} F(U, V) = \sum_{(i,j)\in\Omega} \left((UV^\top)_{ij} - M_{ij}^\star\right)^2, \tag{8}$$

where $r$ is the target rank. If the rank of the ground truth $M^\star$ is known, one can set $r$ accordingly. However, the true rank is often unknown. An alternative approach is to use *overparameterisation* and choose $r$ much higher than needed. **Although in this case $F$ has many minimizers, our results indicate that the gradient flow trajectories (2) for small $\lambda$ tend to converge towards low-rank solutions**.

More precisely, we analyse the *extreme overparameterised setting* when $r = m + n$. In Appendix C, we show that the set $\mathcal{M}^\star$ of stationary points of $F$ which are nonsingular matrices forms a manifold. Provided that unregularised gradient flow converges to a nonsingular point, we can apply our results locally. These results state that, in the second, slow phase of the dynamics, the trajectories minimise $\|U\|_F^2 + \|V\|_F^2$ on $\mathcal{M}^\star$. Recall that for a given matrix $M \in \mathbb{R}^{n \times m}$, we have

$$\|M\|_* = \min_{UV^\top = M} \frac{1}{2}(\|U\|_F^2 + \|V\|_F^2),$$

where $\|M\|_*$ is the nuclear norm [Srebro et al., 2004, Lemma 1]. Since minimising the nuclear norm promotes low-rank solutions, this indicates a drift toward low-rank matrices during the slow phase of the dynamics. In Appendix C, we study the more general class of *matrix sensing problems* and discuss an important technical subtlety: the set $\mathcal{M}^\star$ forms a manifold only after excluding degenerate points. Handling those singularities is highly non-trivial and remains an open direction for future work.

Figure 2 below empirically confirms this grokking for matrix completion. We here randomly generate a rank 3 ground truth matrix $M^\star \in \mathbb{R}^{20 \times 20}$, with non-zero singular values $\sigma_1^\star, \sigma_2^\star, \sigma_3^\star$. We randomly sample 50% of the entries to define the observed entries $\Omega$. We then perform gradient descent with weight decay parameter $\lambda = 10^{-3}$ and stepsize $\gamma = 10^{-2}$ on the loss $F(U, V)$ defined in Equation (8) and where the weights $U, V \in \mathbb{R}^{20 \times 10}$ are initialised with i.i.d. standard Gaussian entries. We then track the training loss, the unmasked test loss $\|M^\star - UV^\top\|_F$, the weight norms $\|w\|^2 = \|U\|_F^2 + \|V\|_F^2$, and singular values of the reconstruction matrix $UV^\top$. Additional experimental details can be found in Appendix D.1.

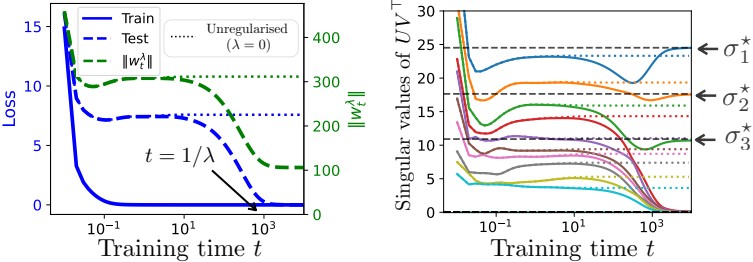

Figure 2: Low-rank matrix completion. *(Left)*: Grokking phenomenon: the training loss drops quickly to zero, while the test loss remains high for an extended period before eventually improving—coinciding with a decrease in the norm of the weights $\|w\|^2 = \|U\|_F^2 + \|V\|_F^2$. *(Right)*: Singular values of $UV^\top$ over time. Each line corresponds to the $i$-th singular value of $UV^\top$. The singular values rapidly converge to large positive values at time $t \approx 1$. However, as grokking starts around time $t \approx 10^2$, all but three begin decay towards zero. The remaining three approach the true singular values $\sigma_1^\star$, $\sigma_2^\star$, and $\sigma_3^\star$.

*Explaining the observed grokking phenomenon.* At time $t = 0$, the weights are randomly initialised and the training loss is high. Initially, the regularised and unregularised weights follow the same trajectory, and the training loss quickly drops to zero: the regularised iterates converge to the same solution as the unregularised gradient flow. This early solution has a high norm, large singular values, and poor generalisation performance. As training continues, around time $t = 1/\lambda$, the weight norms begin to decrease. By $t \approx 10^4$, the parameters have drifted to a new solution that still achieves zero training loss but has a much lower norm and actually coincides with the low rank ground truth matrix $M^\star$.

**Two-layer ReLU network.** Although our theoretical framework does not allow for non-smooth architectures such as ReLU networks, we illustrate in Figure 3 that similar grokking dynamics can be observed in this case. We train a two-layer ReLU network of the form $f_w(x) = \sum_{j=1}^m u_j \operatorname{ReLU}(v_j x + b_j)$, with weights $w = (u, v, b)$ where the outer layer is $u \in \mathbb{R}^m$, the inner weight $v \in \mathbb{R}^m$ and bias $b \in \mathbb{R}^m$. The teacher function $f$ is a sum of 3 ReLUs and is represented in dotted light blue in Figure 3. We generate a training dataset of $n = 10$ points by sampling $x_i$ uniformly in $[-2, 2]$ and computing $y_i = f(x_i)$. These training points are shown as black crosses in Figure 3. We train the student network with $m = 100$ by minimising the squared loss $F(w) = \frac{1}{2n} \sum_{i=1}^n (f_w(x_i) - y_i)^2$ using gradient descent with weight decay $\lambda = 10^{-3}$ for $T = 10^6$ iterations and small step size. The initial weights are independently sampled from a Gaussian of variance 4. At each iteration, we record the train loss, the $\ell_2$-norm of the weights, as well as the test loss over a fixed test dataset (plotted Figure 3, left).

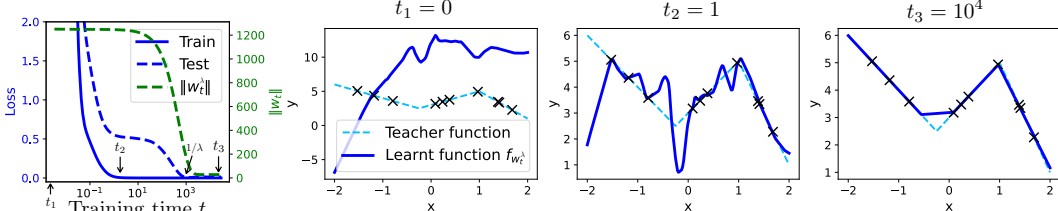

Figure 3: Two-layer ReLU network trained with gradient descent and small weight decay. *(Left)*: Grokking phenomenon: the training loss drops quickly to zero, while the test loss remains high for an extended period before eventually improving—coinciding with a slow, steady decrease in the weight norm. *(Right)*: Snapshots of the network's prediction function at various training times. The ground truth teacher function (a sum of three ReLUs) is shown in dotted light blue, and the training samples are shown as black crosses.

*Explaining the observed grokking phenomenon.* At time $t_1 = 0$, the weights are randomly initialised and the training loss is high. By $t_2 = 1$, the training loss has dropped to nearly zero, and the iterates closely approximate the solution that would be obtained by unregularised gradient flow, this solution does not have a low norm and generalises poorly. Subsequently, around time $t = 1/\lambda$, the weight norms begin to decrease, and by $t_3 \approx 10^5$, they have drifted to a zero training loss solution which has a much lower $\ell_2$-norm and which generalises much better. Such solutions are believed to have a small number of "kinks" [Savarese et al., 2019, Parhi and Nowak, 2021, Boursier and Flammarion, 2023], as observed in Figure 3 (far right plot).

**Diagonal Linear Networks.**   We also study—both as an application of Theorem 1 and numerically—the architecture of *diagonal neural networks* in Appendices C and D.2, which serve as a toy problem for neural network training dynamics. In that case grokking promotes sparse estimators.

## 6   Conclusion

This work presents a rigorous and general optimisation-based description of the grokking phenomenon as a two-timescale process. In the fast initial phase, parameters evolve according to the unregularised flow until reaching a stationary manifold. In the slower second phase, they follow the Riemannian gradient flow of the norm constrained to this manifold. Grokking naturally emerges from a gradual simplicity bias: starting from a poorly generalising solution recovered by unregularised gradient flow, the slow phase driven by weight decay gradually simplifies the model by reducing its norm, ultimately leading to better generalisation.

While prior work has extensively analysed the first phase via the implicit bias of optimisation algorithms, the second phase—norm minimisation constrained to the interpolation manifold—has received little attention. Our framework highlights the critical role of this phase and motivates further study of optimisation dynamics on interpolation manifolds.

Large initialisations (NTK regime) are known to yield poor generalisation [Chizat et al., 2019, Liu et al., 2022c], while small initialisations (rich regime) can lead to slow convergence or convergence to suboptimal solutions for the training loss [Boursier and Flammarion, 2024a,b]. Grokking may offer a desirable compromise, achieving fast convergence to an interpolating solution while retaining strong generalisation.

Note that our analysis is derived in the asymptotic regime $\lambda \to 0$, since this allows for a tractable analysis. Extending the theory to a fixed $\lambda$ is considerably harder, that said, in Appendix E we offer a heuristic analysis regarding how small $\lambda$ needs to be for grokking to emerge. Also note that our analysis can easily be extended to other types of regularisations. In particular, we believe empirical observations reported for training with Sharpness-Aware Minimization [Andriushchenko and Flammarion, 2022, p. 7] may also be interpreted through the lens of grokking, albeit driven by SAM-style regularisation rather than standard weight decay. Lastly, our analysis focuses on regression settings with bounded dynamics. In classification tasks, by contrast, the stationary manifold lies "at infinity" once interpolation is achieved. Extending our approach to such settings remains an open and promising direction for future work, likely requiring techniques tailored to classification losses.

## Acknowledgments and Disclosure of Funding

E. Boursier would like to extend special thanks to Ranko Lazic for his insightful discussions, which were instrumental in initiating this project. S. Pesme would like to thank P. Quinton for carefully reading the paper and providing valuable feedback. R. Dragomir is a chair holder from the Hi! Paris interdisciplinary research center composed of Institut Polytechnique de Paris (IP Paris) and HEC Paris.

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

# Appendix

## Table of Contents

## A    Main proofs

### A.1    Proof of Proposition 1

**Proposition 1.** *If Assumption 1 holds, then for all $T \geq 0$, $w^{\lambda} \underset{\lambda \to 0}{\longrightarrow} w^{GF}$ in $(\mathcal{C}^0([0,T], \mathbb{R}^d), \|\cdot\|_{\infty})$*

*Proof.* We first need to restrict the dynamics of $(w^{\lambda}(t))$ to some compact of $\mathbb{R}^d$. Without loss of generality we can inflate the compact in Assumption 1, so that we can assume there is some compact $K$ of $\mathbb{R}^d$ such that for all $t \geq 0$, $B(w^{\mathrm{GF}}(t), 1) \subset K$, where $B(w^{\mathrm{GF}}(t), 1)$ is the ball of radius 1 centered at $w^{\mathrm{GF}}(t)$. We also define for any $\lambda > 0$, $T_{\lambda} = \inf\{t \in \mathbb{R}_+ \mid w^{\lambda}(t) \notin K\}$.

Thanks to the continuity of $\nabla^2 F$, $\nabla F$ is $c$-Lipschitz on $K$, i.e.,

$$\|\nabla F(w) - \nabla F(w')\| \leq c\|w - w'\| \qquad \text{for any } w, w' \in K.$$

We then derive the following inequalities for all $t \in [0, T_{\lambda})$,

$$
\begin{aligned}
\left\|w^{\lambda}(t) - w^{GF}(t)\right\| &= \left\|\int_0^t \dot{w}^{\lambda}(s) - \dot{w}^{GF}(s)\mathrm{d}s\right\| \\
&= \left\|\int_0^t \nabla F(w^{GF}(s)) - \nabla F(w^{\lambda}(s)) - \lambda w^{\lambda}(s)\mathrm{d}s\right\| \\
&\leq \int_0^t \left\|\nabla F(w^{GF}(s)) - \nabla F(w^{\lambda}(s))\right\|\mathrm{d}s + \lambda \int_0^t \|w^{\lambda}(s)\|\mathrm{d}s \\
&\leq c\int_0^t \left\|w^{\lambda}(s) - w^{GF}(s)\right\|\mathrm{d}s + \lambda t \sup_{w \in K} \|w\|.
\end{aligned}
$$

The integral form of Grönwall inequality[2] – also called Grönwall-Bellman inequality – then leads to the following inequality for any $t \in [0, T_\lambda)$:

$$\left\| w^\lambda(t) - w^{GF}(t) \right\| \leq \lambda t e^{ct} \sup_{w \in K} \|w\|. \tag{9}$$

In particular for a fixed $T \in \mathbb{R}_+$, there is a $\lambda^\star > 0$ small enough such that for any $\lambda \leq \lambda^\star$,

$$\forall t \in [0, T], \lambda t e^{ct} \sup_{w \in K} \|w\| < 1.$$

Given the definition of $T_\lambda$, Equation (9) and the fact that $\bigcup_{t \in \mathbb{R}_+} B(w^{\mathrm{GF}}(t), 1) \subset K$, this then implies that for any $\lambda \leq \lambda^\star$, $T_\lambda > T$. In particular, for any $\lambda \leq \lambda^\star$, Equation (9) becomes

$$\sup_{t \in [0,T]} \left\| w^\lambda(t) - w^{GF}(t) \right\| \leq \lambda T e^{cT} \sup_{w \in K} \|w\|.$$

Proposition 1 then directly follows. $\qquad\square$

## A.2 Junction between fast and slow dynamics

In the limit $\lambda \to 0$, the whole fast dynamics described by Proposition 1 is crushed into the $t = 0$ point of the slow timescale. While Sections 4.1 and 4.2 respectively provide descriptions of the fast and slow timescale dynamics, one needs to control the junction of these two dynamics. This junction is made possible by Lemma 2 below, as well as a timeshift argument in the proof of Proposition 2.

**Lemma 2.** *If Assumption 1 holds, there exists a function $t(\lambda)$ such that $\lim_{\lambda \to 0} t(\lambda) = 0$ and*

$$\lim_{\lambda \to 0} \tilde{w}^\lambda(t(\lambda)) = \lim_{t \to \infty} w^{\mathrm{GF}}(t).$$

Lemma 2 states that for a well chosen timepoint $t(\lambda)$, which is of order $-\lambda \ln(\lambda)$, the slow dynamics solution $\tilde{w}^\lambda$ will be close to the limit of the unregularised flow at that timepoint. This result will then be key in showing that $\lim_{\lambda \to 0} \tilde{w}^\lambda$ admits a right limit in 0, which is given by $\lim_{t \to \infty} w^{\mathrm{GF}}(t)$.

Note that this order $-\lambda \ln(\lambda)$ for $t(\lambda)$ is necessary: a smaller value of $t(\lambda)$ would not allow enough time for the flow to approach the limit of the unregularised gradient flow; while a larger value would correspond to a time where the regularised flow significantly drifted from the unregularised one.

This time $-\lambda \ln(\lambda)$ can indeed be seen, in the fast timescale, as the point where the dynamics transition from mimicking the unregularised flow, to the slow dynamics that minimises the regularisation term within some manifold.

*Proof.* Equation (9) in the proof of Proposition 1 yields that for any $t \in [0, T_\lambda]$, $\left\| w^\lambda(t) - w^{GF}(t) \right\| \leq \lambda t e^{ct} \sup_{w \in K} \|w\|$. We can then define $t(\lambda) = \frac{-\lambda \ln \lambda}{2c} > 0$ and observe that for any $t \leq \min(T_\lambda, \frac{t(\lambda)}{\lambda})$,

$$\|w^\lambda(t) - w^{GF}(t)\| \leq t(\lambda) e^{c \frac{t(\lambda)}{\lambda}} \sup_{w \in K} \|w\|$$

$$= \frac{-\sqrt{\lambda} \ln(\lambda)}{2c} \sup_{w \in K} \|w\|.$$

Since $\sqrt{\lambda} \ln(\lambda) \xrightarrow[\lambda \to 0]{} 0$, we have for $\lambda$ small enough that the above term is smaller than 1. In particular for $\lambda$ small enough, $T_\lambda \geq \frac{t(\lambda)}{\lambda}$. From there, Equation (9) yields at $\frac{t(\lambda)}{\lambda}$ for any small enough $\lambda > 0$

$$\|\tilde{w}^\lambda(t(\lambda)) - \lim_{t \to \infty} w^{GF}(t)\| \leq \|\tilde{w}^\lambda(t(\lambda)) - w^{GF}(\frac{t(\lambda)}{\lambda})\| + \|w^{GF}(\frac{t(\lambda)}{\lambda}) - \lim_{t \to \infty} w^{GF}(t)\|$$

$$\leq \frac{-\sqrt{\lambda} \ln(\lambda)}{2c} \sup_{w \in K} \|w\| + \|w^{GF}(\frac{t(\lambda)}{\lambda}) - \lim_{t \to \infty} w^{GF}(t)\|.$$

Now note that our choice of $t(\lambda)$ is such that both the first and second term in the last inequality converge to 0 — indeed, $t(\lambda) e^{c \frac{t(\lambda)}{\lambda}} = \frac{-\sqrt{\lambda} \ln(\lambda)}{2c} \xrightarrow[\lambda \to 0]{} 0$ and $\frac{t(\lambda)}{\lambda} \xrightarrow[\lambda \to 0]{} +\infty$ — which concludes the proof. $\qquad\square$

---

[2]The more classical form of Grönwall inequality cannot be directly used here, since $\left\| w^\lambda(t) - w^{GF}(t) \right\|$ might be non-differentiable at some points.

### A.3 Proof of Proposition 2

The main element to prove Proposition 2 is Lemma 4 below. We first need to state another auxiliary lemma, given by Lemma 3 below, and proven in Appendix B.5.

**Lemma 3.** *Consider Assumptions 1 and 2. Let $(u^\lambda)_{\lambda>0}$ be a family of solutions of the following ODE for any $\lambda > 0$ and $t \geq 0$,*

$$\dot{u}^\lambda(t) = -u^\lambda(t) - \frac{1}{\lambda}\nabla F(u^\lambda(t)).$$

*If also $u^\lambda(0) \to u_0 \in \mathcal{M}$, then there exists a neighbourhood $U$ of $\mathcal{M}$ such that $\Phi$ is $\mathcal{C}^2$ on $U$ and for every $\varepsilon > 0$, there exists a family of neighbourhoods $(U_\varepsilon)_{\varepsilon>0}$ of $\mathcal{M}$ such that*

1. *$U_\varepsilon \subseteq U_{\varepsilon'} \subset U$ for any $\varepsilon < \varepsilon'$;*

2. *there exists $\lambda(\varepsilon) > 0$ such that for any $\lambda \leq \lambda(\varepsilon)$, the trajectory $(u^\lambda(t))_{t\geq 0}$ is contained in $U_\varepsilon$;*

3. *$\bigcap_{\varepsilon>0} U_\varepsilon = \mathcal{M}$.*

Note that the existence of a neighbourhood $U$ in Lemma 3 is guaranteed by Lemma 1. We can now state our key lemma.

**Lemma 4.** *Consider Assumptions 1 and 2. Let $(u^\lambda)_{\lambda>0}$ be a family of solutions of the following ODE for any $\lambda > 0$ and $t \geq 0$,*

$$\dot{u}^\lambda(t) = -u^\lambda(t) - \frac{1}{\lambda}\nabla F(u^\lambda(t)).$$

*If also $u^\lambda(0) \to u_0 \in \mathcal{M}$, then $u^\lambda$ converges uniformly, as $\lambda \to 0$, on any interval of the form $[0, T]$ to the function $u$ defined as the solution of the following ODE:*

$$u(0) = u_0$$
$$\dot{u}(t) = -D\Phi_{u(t)}(u(t)).$$

*Proof.* First consider a neighbourhood $U$ of $\mathcal{M}$ such that $\Phi$ is $\mathcal{C}^2$ on $U$ and Lemma 3 holds. From there thanks to Lemma 3, we can assume that $\lambda > 0$ is chosen small enough so that $u^\lambda(t) \in U$ for any $t \in \mathbb{R}_+$. We can now compute the time derivative of $\Phi(u^\lambda(t))$, using the chain rule for any $t \in \mathbb{R}_+$ and $\lambda$ small enough:

$$\dot{\Phi}(u^\lambda(t)) = -D\Phi_{u^\lambda(t)} \cdot \left(u^\lambda(t) + \frac{1}{\lambda}\nabla F(u^\lambda(t))\right).$$

Then using the fact that for any $w \in U$, $D\Phi(w) \cdot \nabla F(w) = 0$ [Li et al., 2021, Lemma C.2],

$$\dot{\Phi}(u^\lambda(t)) = -D\Phi_{u^\lambda(t)} \cdot u^\lambda(t). \tag{10}$$

It now remains to show that $u^\lambda(t) \to \mathcal{M}$ as $\lambda \to 0$, to guarantee that $\Phi(u^\lambda(t))$ and $u^\lambda(t)$ have the same limit, which will be done using Lemma 3.

Thanks to Lemma 3, we can consider a family of neighbourhoods $(U_\varepsilon)_{\varepsilon>0}$ of $\mathcal{M}$ and a function $\lambda : \mathbb{R}_+^* \to \mathbb{R}_+^*$ satisfying Lemma 3. As we can always take a smaller choice for any value $\lambda(\varepsilon)$, we can also choose the function $\lambda$ so that

- it is non-decreasing;

- $\lim_{\varepsilon \to 0} \lambda(\varepsilon) = 0$.

For the remaining of proof, define the function $H : w \mapsto -D\Phi_w \cdot w$ and take $\varepsilon(\lambda) = \inf\{\varepsilon > 0 \mid \lambda \leq \lambda(\varepsilon)\}$. We consider in the following $\lambda$ small enough so that $\varepsilon(\lambda)$ is defined and finite. Since $\lim_{\varepsilon \to 0} \lambda(\varepsilon) = 0$, $\varepsilon(\lambda) > 0$ for any $\lambda > 0$. The function $\varepsilon$ is non-increasing, so it admits a limit at 0. Moreover for any $\delta > 0$, $\varepsilon(\lambda(\delta)) \leq \delta$ by definition, so that $\lim_{\lambda \to 0} \varepsilon(\lambda) = 0$.

Thanks to Lemma 3, $(u^\lambda(t))_{t\geq 0}$ is contained in $U_{\varepsilon(\lambda)}$. By monotonicity of $\|u\|_2$, the trajectory $(u(t))_{t\geq 0}$ is bounded. Moreover, the trajectory of $u^\lambda(t)$ is also bounded independently of $\lambda$ thanks to Lemma 7. We can thus consider a compact $K$ of $\mathbb{R}^d$ such that for any $t \geq 0$ and $\lambda > 0$, $u(t) \in K$ and $u(t) \in K$.

Recall that $\Phi$ is the identity function on $\mathcal{M}$ and is $\mathcal{C}^2$ on $U$. In consequence, we have[3] $\sup_{w \in K \cap U_{\varepsilon(\lambda)}} \|\Phi(w) - w\|_2 \xrightarrow[\lambda \to 0]{} 0$.

Summing over Equation (10) yields for any $t \geq 0$:

$$u^\lambda(t) = \int_0^t H(u^\lambda(s))\mathrm{d}s + u^\lambda(t) - \Phi(u^\lambda(t)) + \Phi(u^\lambda(0)).$$

Since $\Phi$ is $\mathcal{C}^2$ on $U$, $H$ is $c$-Lipschitz on $K$, for some $c > 0$. A comparison with $u$ then yields for any $t \geq 0$

$$\|u^\lambda(t) - u(t)\| \leq \int_0^t \|H(u^\lambda(s)) - H(u(s))\|\mathrm{d}s + \|u^\lambda(t) - \Phi(u^\lambda(t))\| + \|\Phi(u^\lambda(0)) - u_0\|$$

$$\leq c \int_0^t \|u^\lambda(s) - u(s)\|\mathrm{d}s + \sup_{w \in K \cap U_{\varepsilon(\lambda)}} \|\Phi(w) - w\| + \|\Phi(u^\lambda(0)) - u_0\|.$$

Similarly to the proof of Proposition 1, an integral form of Grönwall inequality yields for any $t \geq 0$

$$\|u^\lambda(t) - u(t)\| \leq \left( \sup_{w \in K \cap U_{\varepsilon(\lambda)}} \|\Phi(w) - w\| + \|\Phi(u^\lambda(0)) - u_0\| \right) e^{ct}. \tag{11}$$

Noting that the multiplicative term $\sup_{w \in K \cap U_{\varepsilon(\lambda)}} \|\Phi(w) - w\| + \|\Phi(u^\lambda(0)) - u_0\|$ goes to 0 as $\lambda$ goes to 0 allows to conclude on the uniform convergence of $u^\lambda$ to $u$ on $[0, T]$. $\qquad\square$

**Proposition 2.** *If Assumptions 1 and 2 hold, then for all $T, \varepsilon > 0$, we have $\tilde{w}^\lambda \xrightarrow[\lambda \to 0]{} \tilde{w}^\circ$ in $(\mathcal{C}^0([\varepsilon, T], \mathbb{R}^d), \|\cdot\|_\infty)$, where $\tilde{w}^\circ$ is the unique solution on $\mathbb{R}_+$ of the differential equation (6).*

*Proof.* Consider the shifted slow dynamics $\tilde{v}^\lambda$ for any $t \geq 0$ as $\tilde{v}^\lambda(t) = \tilde{w}^\lambda(t + t(\lambda))$ with $t(\lambda)$ given by Lemma 2. Using Lemma 2, $\tilde{v}^\lambda$ then follows the following ODE:

$$\dot{\tilde{v}}^\lambda(t) = -\tilde{v}^\lambda(t) - \frac{1}{\lambda}\nabla F(\tilde{v}^\lambda(t)),$$

with an initial condition satisfying $\lim_{\lambda \to 0} \tilde{v}^\lambda(0) = \Phi(w_0)$.

We can then direct apply Lemma 4 above on $\tilde{v}^\lambda$, which yields that $\tilde{v}^\lambda$ converges uniformly on any interval of the form $[0, T]$ to $\tilde{w}^\circ$.

Proposition 2 is then obtained by observing that $\lim_{\lambda \to 0} t(\lambda) = 0$, so that for any $\varepsilon > 0$ and $\lambda$ small enough such that $t(\lambda) \leq \varepsilon$, it holds for any $t \in [\varepsilon, T]$

$$\|\tilde{w}^\lambda(t) - \tilde{w}^\circ(t)\| = \|\tilde{v}^\lambda(t - t(\lambda)) - \tilde{w}^\circ(t)\|$$

$$\leq \|\tilde{v}^\lambda(t - t(\lambda)) - \tilde{w}^\circ(t - t(\lambda))\| + \|\tilde{w}^\circ(t) - \tilde{w}^\circ(t - t(\lambda))\|.$$

The first term converges to 0 uniformly for $t \in [\varepsilon, T]$ by uniform convergence of $\tilde{v}^\lambda$ towards $\tilde{w}^\circ$; and the second term also goes uniformly to 0 by (uniform) continuity of $\tilde{w}^\circ$ on the considered interval. $\qquad\square$

### A.4 Proof of Proposition 3

**Proposition 3.** *If Assumptions 1 and 2 hold, then for any sequence $(\lambda_k)_{k \in \mathbb{N}}$ such that $\lambda_k \xrightarrow[k \to \infty]{} 0$, the limit points of $(\lim_{t \to \infty} w^{\lambda_k}(t))_{k \in \mathbb{N}}$ are included in the KKT points of Equation (7).*

---

[3]This is a direct consequence of the fact that $\Phi$ is the identity on $\mathcal{M}$, locally Lipschitz and $\lim_{\lambda \to 0} \varepsilon(\lambda) = 0$.

*Proof.* By definition [see e.g., Bergmann and Herzog, 2019], the KKT points of Equation (7) are the points $w^\star \in \mathcal{M}$ satisfying

$$\text{grad}_{\mathcal{M}} \ell_2(w^\star) = 0.$$

Since $\text{grad}_{\mathcal{M}} \ell_2 = P_{\text{Ker}(\nabla^2 F(w^\star))}$, KKT points of Equation (7) are the points $w^\star \in \mathcal{M}$ satisfying

$$w^\star \in \text{Ker}(\nabla^2 F(w^\star))^\perp. \tag{12}$$

Thanks to Lemma 7, the trajectories $(w^\lambda(t))_{t \geq 0}$ are all bounded and $w^\lambda_\infty := \lim_{t \to \infty} w^\lambda(t)$ exists for any $\lambda > 0$. In particular, this limit is a stationary point of the regularised loss $F_\lambda$, i.e.,

$$\nabla F(w^\lambda_\infty) + \lambda w^\lambda_\infty = 0.$$

In particular, $w^\lambda_\infty = -\frac{1}{\lambda} \nabla F(w^\lambda_\infty)$.

Let $(\lambda_k)_{k \in \mathbb{N}}$ be a sequence in $\mathbb{R}^*_+$ such that $\lambda_k \xrightarrow[k \to \infty]{} 0$. Let $w^\star$ be a limit point of the sequence $(w^{\lambda_k}_\infty)_k$. Thanks to Lemma 3, $w^\star \in \mathcal{M}$. Moreover, the equality $w^{\lambda_k}_\infty = -\frac{1}{\lambda_k} \nabla F(w^{\lambda_k}_\infty)$ first implies that $\|\nabla F(w^{\lambda_k}_\infty)\| = \mathcal{O}(\lambda_k)$. Rebjock and Boumal [2024, Proposition 2.8] then also implies that

$$d(w^{\lambda_k}_\infty, \mathcal{M}) = \mathcal{O}(\lambda_k).$$

Moreover, noting $w_k \in \arg\min_{w \in \mathcal{M}} \|w^{\lambda_k}_\infty - w\|$, a Taylor expansion yields

$$\frac{1}{\lambda_k} \nabla F(w^{\lambda_k}_\infty) = \frac{1}{\lambda_k} \nabla F(w_k) + \frac{1}{\lambda_k} \nabla^2 F(w_k)(w^{\lambda_k}_\infty - w_k) + o(\frac{\|w^{\lambda_k} - w_k\|}{\lambda_k})$$

$$= \frac{1}{\lambda_k} \nabla^2 F(w_k)(w^{\lambda_k}_\infty - w_k) + o(\frac{d(w^{\lambda_k}_\infty, \mathcal{M})}{\lambda_k})$$

$$= \frac{1}{\lambda_k} \nabla^2 F(w_k)(w^{\lambda_k}_\infty - w_k) + o(1).$$

The equality $w^{\lambda_k}_\infty = -\frac{1}{\lambda_k} \nabla F(w^{\lambda_k}_\infty)$ then implies for the subsequence $k_n$ associated to the limit point $w^\star$ that

$$-\lim_{n \to \infty} \nabla^2 F(w_{k_n}) \frac{w^{\lambda_{k_n}}_\infty - w_{k_n}}{\lambda_{k_n}} = w^\star. \tag{13}$$

Let $u_k = P_{\text{Ker}(\nabla^2 F(w_k))^\perp} \frac{w^{\lambda_k}_\infty - w_k}{\lambda_k}$. Note that

$$\nabla^2 F(w_k) \frac{w^{\lambda_k}_\infty - w_k}{\lambda_k} = \nabla^2 F(w_k) u_k$$

$$\text{and} \quad \|\nabla^2 F(w_k) u_k\| \geq \eta \|u_k\|,$$

thanks to Assumption 2. In consequence, $(u_k)_k$ is bounded. In particular, it admits an adherence point $u_\infty \in \mathbb{R}^d$.

Since $w_{k_n} \xrightarrow[n \to \infty]{} w^\star$ and $\nabla^2 F$ is continuous, $\|\nabla^2 F(w_{k_n}) - \nabla^2 F(w^\star)\| \xrightarrow[n \to \infty]{} 0$. So that Equation (13) becomes

$$-\nabla^2 F(w^\star) u_\infty = w^\star.$$

In particular, it yields that $w^\star \in \text{Im}(\nabla^2 F(w^\star))$. By symmetry of the Hessian, $\text{Im}(\nabla^2 F(w^\star)) = \text{Ker}(\nabla^2 F(w^\star))^\perp$ so that $w^\star \in \text{Ker}(\nabla^2 F(w^\star))^\perp$, i.e., it satisfies the KKT conditions of Equation (7). $\square$

## A.5 Proof of Proposition 4

**Proposition 4.** *Let Assumptions 1 and 2 hold and, assume additionally that $w^\circ(t)$ converges towards a strict local minimum $w^\star$ of the constrained problem (7). Then $\lim_{\lambda \to 0} \lim_{t \to \infty} w^\lambda(t) = w^\star$.*

*Proof.* For this proof, denote $w^\star = \lim_{t \to \infty} \tilde{w}^\circ(t)$ and $F^\star = F(w^\star)$. $w^\star$ is a strict local minimum of the Euclidean norm on $\mathcal{M}$. Moreover using the Morse Bott property [Assumption 2 and see

Rebjock and Boumal, 2024], we can consider an arbitrarily small $\delta > 0$ such that the following conditions simultaneously hold in $B(0, \delta)$ for some $\beta > 0$:

$$\forall w \in \mathcal{M} \cap B(w^\star, 2\delta), w \neq w^\star \implies \|w^\star\|^2 < \|w\|^2,$$

$$\forall w \in B(w^\star, \delta), F(w) - F^\star \geq \frac{\eta}{4} d(w, \mathcal{M})^2, \tag{14}$$

$$\forall w \in B(w^\star, \delta), \beta(F(w) - F^\star) \geq \|\nabla F(w)\|_2^2 \geq \eta(F(w) - F^\star).$$

First observe that the strict minimality assumption implies, through Lemma 8, that there exists $\varepsilon_0 > 0$ (independent of $\lambda$) such that for a small enough $\lambda > 0$

$$\inf_{\partial B(w^\star, \delta)} F_\lambda(w) > F^\star + \lambda \frac{\|w^\star\|^2}{2} + \lambda \varepsilon_0. \tag{15}$$

Now fix an arbitrarily small $\delta' \in (0, \delta)$. Let $t_0 \in \mathbb{R}_+^*$ such that $\|\tilde{w}^\circ(t_0) - w^\star\| \leq \frac{\delta'}{4}$. By pointwise convergence of $\tilde{w}^\lambda(t_0)$ to $\tilde{w}^\circ(t_0)$, we then have that for $\lambda > 0$ small enough, $\|\tilde{w}^\lambda(t_0) - w^\star\| \leq \frac{\delta'}{2}$. Without loss of generality, we can even choose $t_0$ large enough and $\lambda$ small enough so that for some arbitrarily fixed $\varepsilon > 0$,

$$F_\lambda(\tilde{w}^\lambda(t_0)) \leq F^\star + \varepsilon. \tag{16}$$

From there, we define for this proof $T_\lambda = \inf\{t \geq t_0 \mid \tilde{w}^\lambda(t) \notin B(w^\star, \delta')\}$. Similarly to the proof of Lemma 7, we have for any $t \in (\frac{t_0}{\lambda}, \frac{T_\lambda}{\lambda})$:

$$\frac{\mathrm{d}F_\lambda(w^\lambda(t))}{\mathrm{d}t} \leq -\eta(F_\lambda(w^\lambda(t)) - F^\star) + \lambda(R_1 + \frac{\eta}{2} R^2),$$

where $R_1 = \sup_{w \in B(w^\star, \delta')} \|w\| \|\nabla F(w)\|$ and $R = \sup_{w \in B(w^\star, \delta')} \|w\|$. Again, a Grönwall argument implies that for any $t \in [\frac{t_0}{\lambda}, \frac{T_\lambda}{\lambda}]$,

$$F_\lambda(w^\lambda(t)) \leq F^\star + \varepsilon e^{-\eta(t - \frac{t_0}{\lambda})} + \lambda \left( \frac{R^2}{2} + \frac{R_1}{\eta} \right).$$

In particular, if we define $t' = \min(\frac{t_0}{\lambda} - \frac{\ln(\lambda)}{\eta}, \frac{T_\lambda}{\lambda})$, we have similarly to the proof of Lemma 7 that for any $t \in [\frac{t_0}{\lambda}, t']$:

$$\left\| w^\lambda(t) - w^\lambda(\frac{t_0}{\lambda}) \right\| \leq \beta \sqrt{\varepsilon} \frac{2}{\eta} - C\sqrt{\lambda} \ln(\lambda),$$

for some constant $C$ independent of $\varepsilon$ and $\lambda$. In particular, we can choose $\varepsilon$ and $\lambda$ small enough so that this quantity is smaller than $\frac{\delta'}{2}$. It then implies that $t' < \frac{T_\lambda}{\lambda}$ and

$$F_\lambda(w^\lambda(t')) \leq F^\star + \lambda(\frac{R_1}{\eta} + \frac{R^2}{2} + \varepsilon).$$

From there, by monotonicity of the loss, for any $t \geq t'$:

$$F_\lambda(w^\lambda(t)) \leq F^\star + \lambda(\frac{R_1}{\eta} + \frac{R^2}{2} + \varepsilon).$$

Also note that $\frac{R_1}{\eta} + \frac{R^2}{2} \underset{\delta' \to 0}{\to} \frac{\|w^\star\|^2}{2}$. So we can choose $\delta'$ and $\varepsilon$ small enough so that

$$\frac{R_1}{\eta} + \frac{R^2}{2} + \varepsilon < \frac{\|w^\star\|^2}{2} + \varepsilon_0.$$

From there, the previous inequality implies that for any $t \geq t'$,

$$F_\lambda(w^\lambda(t)) \leq F^\star + \lambda(\frac{\|w^\star\|^2}{2} + \varepsilon_0).$$

By continuity, Equation (15) then implies that for any $t \geq t'$, $w^\lambda(t) \in B(w^\star, \delta)$.

To summarise, we have shown that for any small enough $\delta > 0$, there exists $\lambda^\star(\delta)$ such that for any $\lambda \leq \lambda^\star(\delta)$, $\lim_{t \to \infty} w^\lambda(t) \in B(w^\star, \delta)$.

This means that $\lim_{\lambda \to 0} \lim_{t \to \infty} w^\lambda(t) = w^\star$, which proves Proposition 4. $\qquad \square$

# B Auxiliary proofs

## B.1 Proof of Lemma 1

**Lemma 1.** *If Assumptions 1 and 2 hold, there exists an open neighbourhood $U$ of $\mathcal{M}$ such that $\Phi$ is defined and $\mathcal{C}^2$ on $U$.*

*Proof.* First, we restrict ourselves to a bounded open set $B$ of $\mathbb{R}^d$ and consider $\phi(\cdot, t)$ as a function $B \to \mathbb{R}^d$ for any $t$.

The main point of the proof is to show that $\mathcal{M} \cap B$ is geometrically stable in the sense of Falconer [1983], i.e., that there exists a neighbourhood (in $B$) $U$ of $\mathcal{M} \cap B$, $t > 0$ and $k < 1$ such that for any $w \in U$,

$$d(\phi(w, t), \mathcal{M} \cap B) \leq k d(w, \mathcal{M} \cap B) \qquad \text{and} \qquad \phi(w, t) \in U,$$

where $d(w, \mathcal{M} \cap B) = \inf_{x \in \mathcal{M} \cap B} \|w - x\|_2$.[4]

Let $x \in \mathcal{M} \cap B$. The Morse-Bott property (Assumption 2) implies that there exists a neighbourhood $U(x) \subset B$ of $x$, such that $F$ satisfies the Polyak-Łojasiewicz (PL) inequality with constant $\frac{\eta}{2}$, thanks to the equivalences between both conditions [Rebjock and Boumal, 2024]

$$\|\nabla F(w)\|_2^2 \geq \eta(F(w) - F(x)) \quad \forall w \in U(x). \tag{17}$$

In the following, we define $F^\star = F(x)$, which is the value of $F$ on the manifold $\mathcal{M}$ (the definition does not depend on the choice of $x$).

In particular, there is some $\delta_0(x) > 0$ such that $B(x, \delta_0(x)) \subset U(x)$. Thanks to Rebjock and Boumal [2024, Propositions 2.3 and 2.8, Remark 2.10], we can even choose $\delta_0(x)$ small enough so that there are some $\alpha, \beta$ such that

$$\|\nabla F(w)\|_2 \leq \beta \sqrt{F(w) - F^\star} \quad \text{for any } w \in B(x, \delta_0), \tag{18}$$

$$\frac{\eta}{8} d(w, \mathcal{M})^2 \leq F(w) - F^\star \leq \alpha d(w, \mathcal{M})^2 \text{for any } w \in B(x, \delta_0). \tag{19}$$

By boundedness of $B$, $\alpha$ and $\beta$ can be chosen independently of $x \in \mathcal{M} \cap B$ here.

Now let $w \in B(x, \delta_0(x))$ and define $T(w) = \inf\{t \geq 0 \mid \phi(w, t) \notin B(x, \delta_0(x))\}$. Necessarily, $T(w) > 0$ and for any $t \in [0, T(w))$, Equation (17) applies to $\phi(w, t)$, so that for any $t \in [0, T(w))$

$$\frac{\mathrm{d}(F(\phi(w, t)) - F^\star)}{\mathrm{d}t} = -\|\nabla F(\phi(w, t))\|^2$$
$$\leq -\eta(F(\phi(w, t)) - F^\star).$$

So that, for any $t \in [0, T(w))$:

$$F(\phi(w, t)) - F^\star \leq (F(w) - F^\star)e^{-\eta t}.$$

Moreover for any $t \in [0, T(w))$, Equation (18) also applies, so that

$$\|\phi(w, t) - w\| \leq \int_0^t \|\nabla F(\phi(w, s))\|_2 \mathrm{d}s$$
$$\leq \int_0^t \beta \sqrt{(F(w) - F^\star)e^{-\eta s}} \mathrm{d}s$$
$$\leq \frac{2\beta}{\eta} \sqrt{(F(w) - F^\star)}.$$

By continuity of $F$, let $\delta(x) > 0$ be small enough so that for any $w \in B(x, \delta(x))$, $\delta(x) + \frac{2\beta}{\eta}\sqrt{(F(w) - F^\star)} \leq \frac{\delta_0(x)}{2}$. The previous inequality then implies that for any $w \in B(x, \delta(x))$

---

[4]The definition of Falconer [1983] is stated differently but is implied by our notion of geometric stability, when taking $f(w) = \phi(w, t)$.

and $t \in [0, T(w))$:

$$\|\phi(w,t) - x\| \le \|w - x\| + \|\phi(w,t) - w\|$$
$$\le \delta(x) + \frac{2\beta(x)}{\eta}\sqrt{(F(w) - F^\star)}$$
$$\le \frac{\delta_0(x)}{2}.$$

In particular, for any $w \in B(x, \delta(x))$, $T(w) = \infty$ and $\phi(w,t) \in B(x, \delta_0(x))$ for any $t \ge 0$.

Also, note that $\|\phi(w,t) - x\| \le \frac{\delta_0(x)}{2}$ and $B(x, \delta_0(x)) \subset B$ implies that $d(\phi(w,t), \mathcal{M} \cap B) = d(\phi(w,t), \mathcal{M})$. From there, Equation (19) implies for any $t \ge 0$ and $w \in B(x, \delta(x))$:

$$d(\phi(w,t), \mathcal{M} \cap B)^2 \le \frac{8}{\eta}(F(\phi(w,t)) - F\star)$$
$$\le \frac{8}{\eta}e^{-\eta t}(F(w) - F^\star)$$
$$\le \frac{8\alpha}{\eta}e^{-\eta t}d(w, \mathcal{M} \cap B)^2.$$

In particular, for any $k \ge 0$, we can choose a sufficiently large $t$ such that $d(\phi(w,t), \mathcal{M} \cap B) \le kd(w, \mathcal{M} \cap B)$.

By compactness of $\mathcal{M} \cap B$, there is a finite family of $(x_i)_{i \in [K]} \in \mathcal{M} \cap B$ such that $\bigcup_{i \in [K]} B(x_i, \frac{1}{2}\delta(x_i)) \supseteq \mathcal{M} \cap B$. We then define $U(B) = \bigcup_{i \in [K]} B(x_i, \delta(x_i))$, which is also a finite covering of $\mathcal{M} \cap B$. We then take $t$ large enough such that for any $w \in U(B)$, $d(\phi(w,t), \mathcal{M} \cap B) \le kd(w, \mathcal{M} \cap B)$ for $k < \frac{\min_{i \in [K]} \delta(x_i)}{\max_{i \in [K]} \delta(x_i)}$. In particular, our choice of $k$ is such that, for $f : w \mapsto \phi(w,t)$, $U(B)$ is invariant by $f$. Indeed, note that for any $w \in U(B)$,

$$d(f(w), \mathcal{M} \cap B) \le kd(w, \mathcal{M} \cap B)$$
$$\le \frac{\min_{i \in [K]} \delta(x_i)}{\max_{i \in [K]} \delta(x_i)}d(w, \mathcal{M} \cap B)$$
$$\le \frac{1}{2}\min_{i \in [K]} \delta(x_i).$$

In other words, there is $x \in \mathcal{M} \cap B$ such that $\|f(w) - x\| \le \frac{1}{2}\min_{i \in [K]} \delta(x_i)$. Moreover since $\bigcup_{i \in [K]} B(x_i, \frac{1}{2}\delta(x_i))$ is a covering of $\mathcal{M} \cap B$, there is $j \in [K]$ such that $\|x - x_j\| < \frac{1}{2}\min_{i \in [K]} \delta(x_i)$ and by triangle inequality:

$$\|w - x_j\| < \delta(x_j),$$

i.e., $w \in U(B)$.

Since $U(B)$ is invariant by $f$ and $k < 1$, $\mathcal{M} \cap B$ is geometrically stable, so that we can apply Falconer [1983, Theorem 6.3 and Theorem 5.1]. It then implies that $\Phi$ is $\mathcal{C}^2$ on $U(B)$. Taking an increasing sequence of open bounded sets $B_n$ covering whole $\mathbb{R}^d$, we can then define $U = \bigcup_n U(B_n)$ and conclude that $\Phi$ is $\mathcal{C}^2$ on $U$.

$\square$

## B.2  Minimality of $F$ on neighbourhood

**Lemma 5.** *Consider Assumptions 1 and 2. Let $U$ be an open neighbourhood of $\mathcal{M}$ such that $\Phi$ is continuous on $U$, then necessarily for any $w \in U \setminus \mathcal{M}$, $F(w) = \sup_{x \in \mathcal{M}} F(x)$.*

*Proof.* By definition of $\mathcal{M}$, $F$ is constant on $\mathcal{M}$ so that $\sup_{x \in \mathcal{M}} F(x) = \inf_{x \in \mathcal{M}} F(x) = F^\star$. Moreover $\Phi(\mathcal{M}) = \mathcal{M}$ and by continuity, $\Phi(U) \subset \mathcal{M}$.

For any $w \in U \setminus \mathcal{M}$, note that

$$w - \Phi(w) = -\int_0^\infty \nabla F(\phi(w,t)) \mathrm{d}t,$$

$$F(w) - F^\star = F(\phi(w,0)) - F(\Phi(w)) = -\int_0^\infty \|\nabla F(\phi(w,t))\|^2 \mathrm{d}t.$$

The first equality is the definition of $\Phi(w)$, while the second comes from deriving over time the function $t \mapsto F(\phi(w,t))$ and noting that $\Phi(w) \in \mathcal{M}$.

In particular, for any $w \in U \setminus \mathcal{M}$, $w - \Phi(w) \neq 0$, so that the second integral is also non-zero. In particular, $F(w) > F^\star$ for any $w \in U \setminus \mathcal{M}$. $\qquad\square$

### B.3 Alternative equation for $\tilde{w}^\circ$

**Lemma 6.** *If Assumption 2 holds, the unique solution of Equation (6) also corresponds to the unique solution of the following equation:*

$$\dot{\tilde{w}}^\circ(t) = -D\Phi_{\tilde{w}^\circ(t)}(\tilde{w}^\circ(t)) \quad \text{and} \quad \tilde{w}^\circ(0) = \Phi(w_0).$$

*Proof.* This is a direct consequence of the two following equalities for any $w \in \mathcal{M}$:

$$\mathrm{grad}_{\mathcal{M}} \ell_2(w) = P_{\mathrm{Ker}(\nabla^2 F(w))}(w)$$
$$= D\Phi_w(w).$$

The first one is a consequence of the definition of the Riemannian gradient and the fact that $T_{\mathcal{M}}(w) = \mathrm{Ker}(\nabla^2 F(w))$ [see e.g., Boumal, 2023, Theorem 3.15 with $\nabla F$ being the local defining function of $\mathcal{M}$]. The second one is given by Li et al. [2021, Lemma 4.3]. $\qquad\square$

### B.4 Bounding the trajectories

**Lemma 7.** *If Assumptions 1 and 2 hold, there exists a compact $K$ of $\mathbb{R}^d$ such that for any $\lambda > 0$ and $t \geq 0$, $w^\lambda(t) \in K$. In particular, $\lim_{t\to\infty} w^\lambda(t)$ exists for any $\lambda > 0$.*

*Proof.* Similarly to the proof of Lemma 1, we can consider a neighbourhood $U$ of $\mathcal{M}$ where the PL inequality holds:

$$\|\nabla F(w)\|_2^2 \geq \eta(F(w) - F^\star) \quad \forall w \in U.$$

Additionally, we can consider $\delta > 0$ such that $B(\Phi(w_0), \delta) \subset U$ and

$$\|\nabla F(w)\|_2 \leq \beta\sqrt{F(w) - F^\star} \quad \text{for any } w \in B(\Phi(w_0), \delta).$$

1) For some fixed $\varepsilon > 0$, Lemma 2 then implies there is a $\lambda^\star > 0$ and times $t(\lambda)$ such that for any $\lambda \in (0, \lambda^\star)$ both hold

$$\left\|w^\lambda\left(\frac{t(\lambda)}{\lambda}\right) - \Phi(w_0)\right\| < \frac{\delta}{2} \quad \text{and} \quad F_\lambda\left(w^\lambda\left(\frac{t(\lambda)}{\lambda}\right)\right) - F(\Phi(w_0)) \leq \varepsilon.$$

Moreover, the proof of Lemma 2 also implies that there is some compact $K_1$ of $\mathbb{R}^d$ such that for any $\lambda < \lambda^\star$ and $t \leq \frac{t(\lambda)}{\lambda}$, $w^\lambda(t) \in K_1$.

2) Now fix $\lambda < \lambda^\star$ and define $T_\lambda = \inf\{t \geq \frac{t(\lambda)}{\lambda} \mid w^\lambda(t) \notin B(\Phi(w_0), \delta)\}$. By continuity, $T_\lambda > \frac{t(\lambda)}{\lambda}$. The PL inequality then applies for any $t \in [\frac{t(\lambda)}{\lambda}, T_\lambda)$:

$$\|\nabla F(w^\lambda(t))\|_2^2 \geq \eta(F(w^\lambda(t)) - F^\star).$$

In particular, this allows to derive the following inequalities for any $t \in [\frac{t(\lambda)}{\lambda}, T_\lambda)$:

$$\begin{aligned}
\frac{\mathrm{d}F_\lambda(w^\lambda(t))}{\mathrm{d}t} &= -\|\nabla F_\lambda(w^\lambda(t))\|_2^2 \\
&\leq -\|\nabla F(w^\lambda(t))\|^2 + \lambda\|w^\lambda(t)\|_2\|\nabla F(w^\lambda(t))\|_2 \\
&\leq -\eta(F(w^\lambda(t)) - F^\star) + \lambda R_1 \\
&\leq -\eta(F_\lambda(w^\lambda(t)) - F^\star) + \lambda(R_1 + \frac{\eta}{2}R^2),
\end{aligned}$$

where $R_1 = \sup_{w \in B(\Phi(w_0),\delta)} \|w\| \|\nabla F(w)\|$ and $R = \sup_{w \in B(\Phi(w_0),\delta)} \|w\|$. In particular, Grönwall inequality implies that for any $t \in [\frac{t(\lambda)}{\lambda}, T_\lambda)$

$$F_\lambda(w^\lambda(t)) - F^\star \leq \left( F_\lambda(w^\lambda(\frac{t(\lambda)}{\lambda})) - F^\star \right) e^{-\eta(t - \frac{t(\lambda)}{\lambda})} + \lambda(\frac{R_1}{\eta} + \frac{1}{2}R^2)$$

$$\leq \varepsilon e^{-\eta(t - \frac{t(\lambda)}{\lambda})} + \lambda(\frac{R_1}{\eta} + \frac{1}{2}R^2). \tag{20}$$

Define $t' = \min(\frac{t(\lambda)}{\lambda} + \frac{-\ln(\lambda)}{\eta}, T_\lambda)$. Using Equation (18) for any $t \in (\frac{t(\lambda)}{\lambda}, t']$:

$$w^\lambda(t) - w^\lambda(\frac{t(\lambda)}{\lambda}) = \int_{\frac{t(\lambda)}{\lambda}}^t \nabla F_\lambda(w^\lambda(s)) \mathrm{d}s$$

$$\left\| w^\lambda(t) - w^\lambda(\frac{t(\lambda)}{\lambda}) \right\| \leq \int_{\frac{t(\lambda)}{\lambda}}^{t'} \|\nabla F(w^\lambda(s))\| \mathrm{d}s + \lambda \int_{\frac{t(\lambda)}{\lambda}}^{t'} \|w^\lambda(s)\| \mathrm{d}s$$

$$\leq \beta \int_{\frac{t(\lambda)}{\lambda}}^{t'} \sqrt{F(w^\lambda(s)) - F^\star} \mathrm{d}s - \frac{\lambda \ln(\lambda)}{\eta}(\|\Phi(w_0)\| + \delta).$$

From there, Equation (20) yields for any $t \in (\frac{t(\lambda)}{\lambda}, t']$:

$$\left\| w^\lambda(t) - w^\lambda(\frac{t(\lambda)}{\lambda}) \right\| \leq \beta \int_0^{t' - \frac{t(\lambda)}{\lambda}} \sqrt{\varepsilon} e^{-\frac{\eta}{2}s} \mathrm{d}s + \beta(t' - \frac{t(\lambda)}{\lambda})\sqrt{\lambda(\frac{R_1}{\eta} + \frac{1}{2}R^2)} - \frac{\lambda \ln(\lambda)}{\eta}(\|\Phi(w_0)\| + \delta)$$

$$\leq \beta\sqrt{\varepsilon}\frac{2}{\eta} - C\sqrt{\lambda}\ln(\lambda) - \frac{\lambda \ln(\lambda)}{\eta}(\|\Phi(w_0)\| + \delta),$$

for some constant $C$, which is independent of both $\varepsilon$ and $\lambda$. In particular, we can choose $\varepsilon$ and $\lambda^\star$ small enough, so that $\|w^\lambda(t) - w^\lambda(\frac{t(\lambda)}{\lambda})\| < \frac{\delta}{2}$ for any $t \in (\frac{t(\lambda)}{\lambda}, t']$. By definition, this implies that $t' < T_\lambda$, i.e., for any $t \in [\frac{t(\lambda)}{\lambda}, t']$, $w^\lambda(t) \in B(\Phi(w_0), \delta)$.

3) Since $t' < T_\lambda$, $t' = \frac{t(\lambda)}{\lambda} + \frac{-\ln(\lambda)}{\eta}$ by definition and

$$F_\lambda(w^\lambda(t')) \leq F^\star + \lambda(\frac{R_1}{\eta} + \frac{1}{2}R^2 + \varepsilon).$$

By monotonicity of the objective, we then have for any $t \geq t'$:

$$F_\lambda(w^\lambda(t)) \leq F^\star + \lambda(\frac{R_1}{\eta} + \frac{1}{2}R^2 + \varepsilon). \tag{21}$$

Now define $\tilde{T}_\lambda = \inf\left\{ t \geq \frac{t(\lambda)}{\lambda} \mid w^\lambda(t) \notin U \right\}$. Since $\mathcal{M}$ minimizes $F$ on $U$, Equation (21) implies by continuity that for any $t \in [t', \tilde{T}_\lambda]$:

$$\frac{1}{2}\|w^\lambda(t)\|^2 \leq \frac{R_1}{\eta} + \frac{1}{2}R^2 + \varepsilon.$$

In particular, for $K_2 = B(0, \frac{2R_1}{\eta} + R^2 + 3\varepsilon)$ and any $t \in [t', \tilde{T}_\lambda]$, $w^\lambda(t) \in K_2$. By continuity and compactness, $\inf_{w \in (\partial U) \cap K_2} F(w) > F^\star$ thanks to Lemma 5. In consequence, we can choose $\lambda$ small enough so that Equation (21) implies that for any $t \in [t', \tilde{T}_\lambda]$,

$$F(w^\lambda(t)) < \inf_{w \in (\partial U) \cap K_2} F(w).$$

Assume now that $\tilde{T}_\lambda < \infty$. Since $w^\lambda(\tilde{T}_\lambda) \in K_2$, the previous inequality implies by continuity that $w^\lambda(\tilde{T}_\lambda) \notin \partial U$, i.e., $w^\lambda(\tilde{T}_\lambda) \in \mathring{U}$. This however contradicts the definition of $\tilde{T}_\lambda$, so that $\tilde{T}_\lambda = \infty$. In particular for any $t \geq t'$, $w^\lambda(t) \in K_2$.

To summarize, we have showed that there exists a small enough $\lambda^\star$, such that for any $\lambda \leq \lambda^\star$:

1. $w^\lambda(t)$ is included in some compact $K_1$ of $\mathbb{R}^d$ for $t \leq \frac{t(\lambda)}{\lambda}$;

2. $w^\lambda(t)$ is included in $B(\Phi(w_0), \delta)$ or $t \in (\frac{t(\lambda)}{\lambda}, t')$;

3. $w^\lambda(t)$ is included in some compact $K_2$ of $\mathbb{R}^d$ for $t \geq t'$;

where $K_1$ and $K_2$ are both independent of $\lambda$. In particular, there exists a compact $K$ of $\mathbb{R}^d$ independent of $\lambda$ such that for any $\lambda \leq \lambda^\star$, the trajectory of $(w^\lambda(t))_{t\geq 0}$ is included in $K$.

For $\lambda \geq \lambda^\star$, we directly have by monotonicity of the objective that for any $t \geq 0$

$$\frac{1}{2}\|w^\lambda(t)\|^2 \leq \frac{1}{2}\|w^\lambda(0)\|^2 + \frac{1}{\lambda}\left(F(w^\lambda(0)) - F(w^\lambda(t))\right)$$
$$\leq \frac{1}{2}\|w^\lambda(0)\|^2 + \frac{1}{\lambda^\star}F(w^\lambda(0)),$$

so that the trajectory $(w^\lambda(t))_{t\geq 0}$ is also included in a compact independent of $\lambda$.

As a consequence, the definability assumption of $F_\lambda$ along with the boundedness implies that $\lim_{t\to\infty} w^\lambda(t)$ exists thanks to Kurdyka [1998, Theorem 2]. $\qquad\square$

## B.5  Proof of Lemma 3

**Lemma 3.** *Consider Assumptions 1 and 2. Let $(u^\lambda)_{\lambda>0}$ be a family of solutions of the following ODE for any $\lambda > 0$ and $t \geq 0$,*

$$\dot{u}^\lambda(t) = -u^\lambda(t) - \frac{1}{\lambda}\nabla F(u^\lambda(t)).$$

*If also $u^\lambda(0) \to u_0 \in \mathcal{M}$, then there exists a neighbourhood $U$ of $\mathcal{M}$ such that $\Phi$ is $\mathcal{C}^2$ on $U$ and for every $\varepsilon > 0$, there exists a family of neighbourhoods $(U_\varepsilon)_{\varepsilon>0}$ of $\mathcal{M}$ such that*

1. *$U_\varepsilon \subseteq U_{\varepsilon'} \subset U$ for any $\varepsilon < \varepsilon'$;*

2. *there exists $\lambda(\varepsilon) > 0$ such that for any $\lambda \leq \lambda(\varepsilon)$, the trajectory $(u^\lambda(t))_{t\geq 0}$ is contained in $U_\varepsilon$;*

3. *$\bigcap_{\varepsilon>0} U_\varepsilon = \mathcal{M}$.*

*Proof.* We consider the neighbourhood $U$ defined as in Lemma 7. By definition, $F$ is constant on the manifold $\mathcal{M}$ and denote its value $F^\star$, i.e., $F^\star = \sup_{w\in\mathcal{M}} F(w) = \inf_{w\in\mathcal{M}} F(w)$.

For any $\varepsilon > 0$, we define $U_\varepsilon$ as $U_\varepsilon = \{w \in U \mid F(w) < F^\star + \varepsilon\}$ and show that it satisfies these three conditions. By continuity of $F$, $U_\varepsilon$ is a neighbourhood of $\mathcal{M}$ and the first condition is obviously satisfied.

Thanks to Lemma 5, for any $w \in U \setminus \mathcal{M}$, $F(w) > F^\star$. This implies the third condition,

$$\bigcap_{\varepsilon>0} U_\varepsilon = \mathcal{M}.$$

The arguments of Lemma 7 extend to any family of solutions $(u^\lambda)_{\lambda>0}$ satisfying the assumptions of Lemma 3. In consequence, we can consider a compact $K$ of $\mathbb{R}^d$ such that for any $t \geq 0$ and $\lambda > 0$, $u^\lambda(t) \in K$. From there, note again that $\inf_{w\in(\partial U)\cap K} F(w) > F^\star$.

Since $u^\lambda(0) \to u_0 \in \mathcal{M}$, we can then choose $\lambda(\varepsilon) > 0$ small enough, so that for any $\lambda \in (0, \lambda(\varepsilon)]$,

$$F(u^\lambda(0)) + \frac{\lambda}{2}\|u^\lambda(0)\|_2^2 < \min(F^\star + \varepsilon, \inf_{w\in(\partial U)\cap K} F(w)).$$

By monotonicity of the objective over time, $F(u^\lambda(t)) + \lambda\|u^\lambda(t)\|_2^2 < \min(F^\star + \varepsilon, \inf_{w\in(\partial U)\cap K} F(w))$ for any $t \geq 0$. Since $u^\lambda(t)$ is continuous, it implies that $u^\lambda(t) \in \overset{\circ}{U}_\varepsilon$ for any $t \geq 0$, which concludes the proof of Lemma 3. $\qquad\square$

## B.6 Strict Minimality

**Lemma 8.** *Under the same assumptions than Proposition 4 with $w^\star = \lim_{t\to\infty} \tilde{w}^\circ(t)$, there exists a $\delta^\star > 0$ such that for any $\delta \in (0, \delta^\star)$, there exists $\varepsilon > 0$ and $\lambda^\star > 0$ such that for any $\lambda \in (0, \lambda^\star)$*

$$\inf_{\partial B(w^\star, \delta)} F_\lambda(w) > F^\star + \lambda \frac{\|w^\star\|^2}{2} + \lambda \varepsilon. \tag{22}$$

*Proof.* Let $\delta^\star > 0$ be such that for any $w \in \mathcal{M} \cap B(w^\star, 2\delta^\star)$, $w \neq w^\star \implies \|w\|^2 > \|w^\star\|^2$ and such that Equation (14) holds. Now let $\delta \in (0, \delta^\star)$. We now fix $\lambda^\star > 0$ arbitrarily small, choose $\lambda \in (0, \lambda^\star)$ and define

$$\varepsilon = \frac{1}{4} \inf_{\substack{u \in \mathcal{M} \\ \frac{\delta}{2} \leq \|u - w^\star\| \leq 2\delta}} \|u\|_2^2 - \|w^\star\|_2^2.$$

By strict minimality, compactness and continuity, $\varepsilon > 0$.

Let now $w \in \partial B(w^\star, \delta)$. We can decompose $w$ as $w = u + v$, where $u \in \arg\min_{w' \in \mathcal{M}} \|w - w'\|$. Necessarily, $\|v\| = d(w, \mathcal{M}) \leq \delta$ and $\|w - w^\star\| = \delta$. In particular, we also have $2\delta \geq \|u - w^\star\| \geq \delta - \|v\|$. From there, using the quadratic growth property (Equation 14):

$$\begin{aligned}
F_\lambda(w) &\geq F(w) + \frac{\lambda}{2}(\|u\| - \|v\|)^2 \\
&\geq F^\star + \frac{\eta}{4} d(w, \mathcal{M})^2 + \frac{\lambda}{2}(\|u\|^2 - 2\|u\|\|v\|) \\
&\geq F_\lambda(w^\star) + \frac{\eta}{4} d(w, \mathcal{M})^2 - \lambda(\|w^\star\| + 2\delta)\|v\| + \frac{\lambda}{2}(\|u\|^2 - \|w^\star\|^2).
\end{aligned}$$

Let $c(\varepsilon, \delta) = \min(\frac{\delta}{2}; \frac{\varepsilon}{\|w^\star\| + 2\delta}) > 0$. There are two cases.

1) Either $\|v\| \leq c(\varepsilon, \delta)$, in which case $2\delta \geq \|u - w^\star\| \geq \frac{\delta}{2}$, so that by definition of $\varepsilon$

$$\begin{aligned}
F_\lambda(w) &\geq F_\lambda(w^\star) - \lambda(\|w^\star\| + 2\delta)\|v\| + \frac{\lambda}{2}(\|u\|^2 - \|w^\star\|^2) \\
&\geq F_\lambda(w^\star) - \lambda(\|w^\star\| + 2\delta)c(\varepsilon, \delta) + \frac{\lambda}{2} \cdot 4\varepsilon \\
&\geq F_\lambda(w^\star) + \lambda \varepsilon.
\end{aligned}$$

2) Or $\|v\| \geq c(\varepsilon, \delta)$, in which case we simply have, also using that $\|v\| \leq \delta$:

$$\begin{aligned}
F_\lambda(w) &\geq F_\lambda(w^\star) + \frac{\eta}{4}\|v\|^2 - \lambda(\|w^\star\| + 2\delta)\|v\| \\
&\geq F_\lambda(w^\star) + \frac{\eta}{4}c(\varepsilon, \delta)^2 - \lambda(\|w^\star\| + 2\delta)\delta.
\end{aligned}$$

In particular, choosing $\lambda^\star$ small enough – depending on $\eta, \varepsilon$ and $\delta$ – we have for any $\lambda \leq \lambda^\star$ that

$$\frac{\eta}{4}c(\varepsilon, \delta)^2 - \lambda(\|w^\star\| + 2\delta)\delta \geq \lambda \varepsilon.$$

So that in both cases, $F_\lambda(w) > F^\star + \lambda \frac{\|w^\star\|^2}{2} + \lambda \varepsilon$. $\qquad\square$

# C  Applications

In this section, we provide additional details to the examples discussed in Section 5, and specify how our theoretical results can be applied in various settings.

**Linear regression.**  We consider $F(w) = \|Xw - y\|_2^2$ with $X \in \mathbb{R}^{n \times d}$ and $n \leq d$; assume for simplicity that $X$ is full rank. In this setting, the dynamics can be computed explicitely to illustrate our result.

Denote the solution of minimal $\ell_2$ norm with $w^\star = X^+ y$, where $X^+$ is the Moore-Penrose pseudoinverse of $X$. The problem is convex and the critical set of $F$ is the affine subspace $\mathcal{M} = w^\star + \mathrm{Ker}(X)$, which is a manifold: Assumption 2 is satisfied.

Consider the singular value decomposition $X = U\Sigma V^\top$ where $U \in \mathbb{R}^{n \times d}, V \in \mathbb{R}^{d \times d}$ are orthogonal and $\Sigma = \mathrm{diag}(\sigma_1, \ldots, \sigma_d)$ with $\sigma_{n+1} = \cdots = \sigma_d = 0$. We make the change of coordinates $z = V^\top w$, and notice that in this basis the minimum norm solution $z^\star = V^\top w^\star$ is of the form $z^\star = (z_1^\star, \ldots z_n^\star, 0, \ldots, 0)$. Then, we can compute the trajectory of the gradient flow on $F_\lambda$ initialized at $z(0) = V^\top w_0$:

- for $1 \leq i \leq n$,

$$z_i^\lambda(t) = z_i^{\lambda,\infty} + e^{-(\sigma_i^2 + \lambda)t}\left(z_i(0) - z_i^{\lambda,\infty}\right) \quad \text{with} \quad z_i^{\lambda,\infty} = \frac{\sigma_i^2}{\sigma_i^2 + \lambda}z_i^\star, \qquad (23)$$

- for $(n+1) \leq i \leq d$,

$$z_i^\lambda(t) = e^{-\lambda t}z_i(0). \qquad (24)$$

Eq. (23) describes the dynamics along the directions **orthogonal to** $\mathcal{M}$, and Eq. (24) along those **parallel to** $\mathcal{M}$. When $\lambda \to 0$, the first is much faster than the second. In the first phase, the iterates converge to $(z_1^{\lambda,\infty}, \ldots z_n^{\lambda,\infty}, z_{n+1}(0), \ldots, z_d(0)) \overset{\lambda \to 0}{\approx} (z_1^\star, \ldots z_n^\star, z_{n+1}(0), \ldots, z_d(0))$; this is the limit of unregularised gradient flow $z^{\mathrm{GF}}$ (which is also here the projection of the initial point onto $\mathcal{M}$). In the second phase, the iterates converge slowly towards the mimimum norm solution $(z_1^\star, \ldots z_n^\star, 0, \ldots, 0)$.

**Diagonal linear networks (DLNs).**  DLNs serve as a toy example to understand the influence of the architecture on the training dynamics of neural networks [Pesme, 2024]. The corresponding optimization problem writes

$$\min_{(w_1, w_2) \in \mathbb{R}^{2d}} \|X(w_1 \odot w_2) - y\|_2^2,$$

where $\odot$ denotes the componentwise product, and $X \in \mathbb{R}^{n \times d}$ is the feature matrix with $n \leq d$, which we assume to be full rank. It is usually convenient to perform a rotation of the coordinates and rewrite the problem as

$$\min_{(u,v) \in \mathbb{R}^{2d}} F(u,v) = \|X(u^2 - v^2) - y\|_2^2,$$

where $u^2, v^2$ denotes the componentwise square. The critical set of $F$ is composed of the couples $(u,v)$ satisfying

$$\begin{aligned}
u \odot \left[X^\top(X(u^2 - v^2) - y)\right] &= 0, \\
v \odot \left[X^\top(X(u^2 - v^2) - y)\right] &= 0
\end{aligned} \qquad (25)$$

This set has singularities for points who have null coordinates; if we exclude those problematic points, we can show that it is a manifold.

**Proposition 5.** *The set $\mathcal{M}^* = \nabla F^{-1}(0) \cap (\mathbb{R}^*)^{2d}$ is a smooth manifold of dimension $2d - n$.*

*Proof.* Let $(\bar{u}, \bar{v}) \in \mathcal{M}^*$. Denote $W$ a neighborhood of $(\bar{u}, \bar{v})$ such that $U \subset (\mathbb{R}^*)^{2d}$. The function $H : \mathbb{R}^{2d} \to R^d$ with $H(u,v) = X^\top(X(u^2 - v^2) - y)$ is a local defining function for $\mathcal{M}^*$, in the sense that for $(u,v) \in W$, we have $(u,v) \in \mathcal{M}^* \iff H(u,v) = 0$.

The differential of $H$ at $(\bar{u}, \bar{v})$ is the linear map satisfying for $(\Delta u, \Delta v) \in \mathbb{R}^{2d}$

$$DH(\bar{u}, \bar{v})[\Delta u, \Delta v] = 2X^\top X(\bar{u} \odot \Delta u - \bar{v} \odot \Delta v).$$

It is clear that, since all coordinates of $(\bar{u}, \bar{v})$ are nonzero, the map $(\Delta u, \Delta v) \mapsto \bar{u} \odot \Delta u - \bar{v} \odot \Delta v$ is a surjection on $\mathbb{R}^d$, and therefore $\mathrm{rank}(DH(\bar{u}, \bar{v})) = \mathrm{rank}(X^\top X) = n$. This proves that $\mathcal{M}^*$ is a manifold of dimension $2d - n$ [Boumal, 2023, §3.2]. $\qquad\square$

Because of the singular points in $\mathcal{M}$, the function $F$ does not satisfy Assumption 2 globally. However, our results can still be applied *locally*: see the paragraph below for details.

Noting that, for a vector $w \in \mathbb{R}^d$, we have

$$\|w\|_1 = \min_{u,v \in \mathbb{R}^d} \|u\|_2^2 + \|v\|_2^2 \ \text{ subject to } \ u^2 - v^2 = w,$$

we conclude that in the second, slow phase of the dynamics, the Riemannian gradient flow which minimizes $\|u\|^2 + \|v\|^2$ on $\mathcal{M}^*$ tends to drift towards solutions of low $\ell_1$ norm.

**Low-rank matrix sensing/completion.** Let $\mathcal{A} : \mathbb{S}^n \to \mathbb{R}^m$ be a linear map on symmetric matrices with $m \le n^2$ and $y \in \mathbb{R}^m$. For a given target rank $r \le n$, the matrix sensing problem is

$$\min_{W \in \mathbb{R}^{n \times r}} F(W) = \|\mathcal{A}(WW^\top) - y\|_2^2 \tag{26}$$

A typical example is symmetric matrix completion, where the goal is to recover an unknown matrix $M^* \in \mathbb{R}^{n \times n}$ from a subset of observed entries with coefficients in $\Omega \in \{1 \ldots n\}^2$: the objective function writes $F(W) = \sum_{(i,j) \in \Omega} \left( (WW^\top)_{ij} - M^*_{ij} \right)^2$. Note that the asymmetric case presented in Section 5, Equation (8), can also be written as a symmetric matrix completion problem, by setting

$$W = \begin{bmatrix} U \\ V \end{bmatrix} \in \mathbb{R}^{(n+m) \times r},$$

and choosing a new mask $\Omega$ that selects only the off-diagonal blocks of $WW^\top$.

Usually, one looks for a low-rank solution to Problem (26), by setting $r$ to a small value. Here, we choose to rather study the **overparameterised** setting where $r = n$. Our results imply that, even though we do not explicitly impose a low rank structure, the gradient flow trajectories $W^\lambda$ are driven towards a low-rank solution in the second phase of the dynamics.

Similarly to the example of diagonal linear networks, we show that, in the overparameterised setting, the critical set of $F$ is a manifold if we exclude singular matrices.

**Proposition 6.** *Let $F$ be the matrix sensing function defined in* (26)*, and denote $\mathbb{R}^{n \times n}_*$ the set of invertible matrices of size $n \times n$. If $r = n$, the set $\mathcal{M}^* = \nabla F^{-1}(0) \cap \mathbb{R}^{n \times n}_*$ is a smooth manifold.*

*Proof.* The gradient of $F$ is

$$\nabla F(W) = 4\mathcal{A}^* \left( \mathcal{A}(WW^\top) - y \right) W, \quad \forall W \in \mathbb{R}^{n \times n},$$

where $\mathcal{A}^* : \mathbb{R}^m \to \mathbb{S}^n$ is the adjoint of $\mathcal{A}$.

Let $\overline{W} \in \mathcal{M}^*$, and let $\mathcal{U}$ a neighborhood of $\overline{W}$ such that $\mathcal{U} \subset \mathbb{R}^{n \times n}_*$. For $W \in \mathcal{U}$, $W$ is invertible and we have $W \in \mathcal{M}^*$ if and only if $\mathcal{A}^*(\mathcal{A}(WW^\top) - y) = 0$. The function $H(W) = \mathcal{A}^*(\mathcal{A}(WW^\top) - y)$ is therefore a local defining function for $\mathcal{M}^*$. Its differential at $\overline{W}$ satisfies for $U \in \mathbb{R}^{n \times n}$,

$$DH(\overline{W})[U] = \mathcal{A}^* \mathcal{A}(\overline{W}U^\top + U\overline{W}^\top).$$

Since $\overline{W}$ is invertible, the map $\phi : U \mapsto \overline{W}U^\top + U\overline{W}^\top$ is a surjection from $\mathbb{R}^{n \times n}$ onto $\mathbb{S}^n$: indeed, note that for any $Z \in \mathbb{S}^n$, we have $\phi(U) = Z$ with $U = \frac{1}{2} Z(\overline{W}^{-1})^\top$. Therefore, the rank of $DH(\overline{W})$ is equal to the rank of $\mathcal{A}^* \mathcal{A}$ for any $\overline{W} \in \mathcal{M}^*$, which proves that $\mathcal{M}^*$ is a smooth manifold. $\qquad\square$

**Dealing with singularities.** In the last two examples, the set $\nabla F^{-1}(0)$ has singular points, and so Assumption 2 does not hold globally. However, we showed that it holds on "most of the space", as there exists a negligible set $\mathcal{S}$ such that $\mathcal{M}^* = \nabla F^{-1}(0) \setminus \mathcal{S}$ is a smooth manifold.

Our results can still be applied locally, assuming that the unregularised gradient flow $w^{\mathrm{GF}}$ converges to a point $w^{\mathrm{GF}}_\infty \in \mathcal{M}^*$. Indeed, in that case there exists a neighborhood $\mathcal{U}$ of $w^{\mathrm{GF}}_\infty$ such that $\nabla F^{-1}(0) \cap \mathcal{U}$ is included in $\mathcal{M}^*$. Then, the Morse-Bott property holds in this neighborhood.

Consider then the Riemannian gradient flow $w^\circ$ of the $\ell_2$ norm on $\mathcal{M}^*$ initialized at $w_\infty^{\mathrm{GF}}$. For any time horizon $T$ such that the trajectory of $w^\circ$ stays in $\mathcal{U}$ on the interval $[0, T]$, we can restrict our analysis to this local region, where our assumptions are satisfied. We can then invoke Proposition 2 to conclude that $\tilde{w}^\lambda$ converges to $w^\circ$ uniformly on intervals of the form $[\epsilon, T]$.

However, a key limitation arises when analyzing the long-time behavior: the results characterizing the limit points ( Proposition 4) do not apply if $w^\circ$ converges to a singular point outside $\mathcal{M}^*$. This situation can occur, as singular points might correspond to points that minimize the $\ell_2$ norm on $\mathcal{M}^*$ (e.g., sparse vectors for diagonal networks, or low-rank matrices for matrix sensing). Establishing convergence of $w^\lambda$ to such singular points remains an open and challenging problem, which we leave for future work.

In summary, **our results capture the grokking dynamics near nonsingular points in $\mathcal{M}^*$, but do not yet account for potential convergence toward singular points, which represents an important open challenge.**

# D    Additional experiments

## D.1    Additional Experimental Details

In all our figures and to align with the continuous-time analysis, training iterations refers to the rescaled "training time" $t_k = \gamma k$, where $k$ is the number of gradient steps and $\gamma$ the gradient descent stepsize. We run gradient descent $10^7$ iterations for Figure 2 and $10^6$ iterations for Figure 3.

## D.2    Diagonal linear networks.

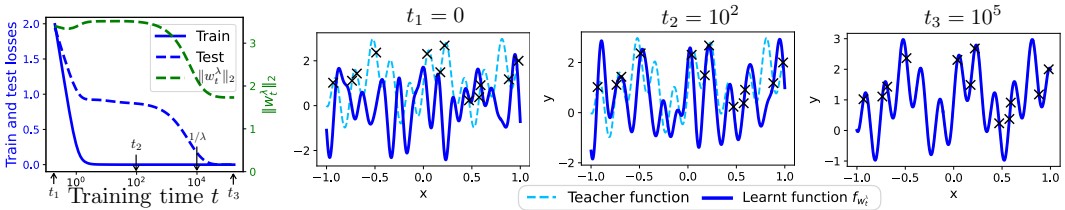

Figure 4: Gradient flow with small weight decay $\lambda$ on a two-layer diagonal linear network. Regression dataset. *(Left)*: Empirical observation of the grokking behaviour. The training loss rapidly drops to zero, while the test loss remains flat for an extended period before eventually decreasing. This transition coincides with a slow but steady decrease in the $\ell_2$-norm of the weights. *(Three plots on the right)*: Visualisation of the model predictions throughout training. The dotted light blue curve represents the teacher function, and the crosses indicate the training data. Snapshots of the model's prediction function at various training times (shown in increasing colour intensity) illustrate how generalisation is affected before and after the transition at $t \approx 1/\lambda$.

*Experimental setup (Figure 4).* We train a two-layer diagonal linear network of the form $f_w(x) = \langle u \odot v, \varphi(x) \rangle$, where $w = (u, v) \in \mathbb{R}^{2d}$ and $\odot$ denotes element-wise multiplication, on a 1D toy dataset. The input $x \in \mathbb{R}$ is mapped to a high-dimensional feature space via the feature map $\varphi(x) = \left[ 1, \cos\left(\frac{\pi x}{2}\right), \ldots, \cos\left(\frac{\pi d_f x}{2}\right), \sin\left(\frac{\pi x}{2}\right), \ldots, \sin\left(\frac{\pi d_f x}{2}\right) \right]$, with $d_f = 30$. The teacher function is a sparse Fourier series $f(x) = 1 + \cos\left(\frac{6\pi x}{2}\right) + \sin\left(\frac{21\pi x}{2}\right)$ and is shown as a dotted light blue curve in Figure 4. The training dataset consists of $n = 12$ input-output pairs $(x_i, y_i)$, where $x_i$ are sampled uniformly in $[-1, 1]$ and $y_i = f(x_i)$. These training points are shown as crosses in Figure 4. We optimise the squared loss $F(w) = \frac{1}{2n} \sum_{i=1}^n (y_i - f_w(x_i))^2$ using gradient descent with weight decay $\lambda = 10^{-4}$. Finally, the initial weights are sampled from a centered Gaussian of variance $0.1$.

*Explaining the observed grokking phenomenon.* At time $t_1 = 0$, the weights are randomly initialised and the training loss is high. By $t_2 = 10^2$, the training loss has dropped to nearly zero, and the iterates closely approximate the solution that would be obtained by unregularised gradient flow. This solution is fully characterised by the implicit regularisation result of [Woodworth et al., 2020], and it does not have a low norm.[5] Subsequently, around time $t = 1/\lambda$, the weight norms begin to decrease, and by $t_3 \approx 10^5$, they converge to the minimum-norm solution $(u^\star, v^\star) = \arg\min_{F(u,v)=0} \|u\|_2^2 + \|v\|_2^2$. A straightforward calculation shows that the elementwise product $\beta^\star := u^\star \odot v^\star$ solves the problem $\arg\min_{\langle \beta, x_i \rangle = y_i \forall i} \|\beta\|_1$. This is an $\ell_1$-minimisation problem, which (under RIP conditions) is known to recover the sparsest solution [Candes, 2008], explaining the zero test loss after the grokking phenomenon.

---

[5]One could also reach the solution observed at time $t_3 = 10^5$ without using weight decay by employing a much smaller initialisation scale [Woodworth et al., 2020], but at the cost of longer training time.

# E   Heuristic analysis on how small $\lambda$ needs to be for grokking to emerge

Note that our theoretical results are derived in the asymptotic regime $\lambda \to 0$, since this setting allows for a tractable and general analysis. Extending the theory to obtain explicit results for a fixed $\lambda > 0$ is considerably more challenging. That said, we can offer some intuition regarding how small $\lambda$ needs to be for grokking to emerge.

Grokking depends on a clear separation between two phases: an initial phase where the iterates converge and stagnate at the solution of the unregularised gradient flow, and a second phase driven by weight decay, during which test performance improves. For grokking to be observable, the regularised gradient flow should approach the unregularised limit before weight decay begins to significantly influence the dynamics.

To formalize this intuition, we can define two characteristic times: $t_{\mathrm{GF}}$, the convergence time of the unregularized gradient flow, measured as the time at which the gradient norm substantially decreases relative to its initial value; and a second time $t_{\mathrm{WD}}$ of order $1/\lambda$, associated with the onset of the regularization effects. When $\lambda$ is small enough that $t_{\mathrm{GF}} \ll 1/\lambda$, we expect to observe grokking-like behavior. Specifically, for some threshold $\varepsilon \ll 1$ (e.g., $\varepsilon = 0.01$): let $t_{\mathrm{GF}}$ such that $\|\nabla F(w_{t_{\mathrm{GF}}})\| \approx \varepsilon \|\nabla F(w_0)\|$. Now let $t_{\mathrm{WD}}$ denote the time when weight decay kicks in: i.e. when the magnitude of the unregularised gradient becomes comparable to the magnitude of the weight decay term: $\|\nabla F(w_{t_{\mathrm{WD}}})\| \approx \lambda \|w_{t_{\mathrm{WD}}}\|$. Since at time $t_{\mathrm{WD}}$, the solution is close to the gradient flow solution $w^{\mathrm{GF}}$, we can consider $\|w_{t_{\mathrm{WD}}}\| \approx \|w^{\mathrm{GF}}\|$. The condition for grokking to occur (i.e., a plateau in test loss followed by an improvement of the test loss) is thus that $t_{\mathrm{GF}} \ll t_{\mathrm{WD}}$. Translating this condition in terms of gradients, we obtain: $\|\nabla F(w_{t_{\mathrm{GF}}})\| \gg \|\nabla F(w_{t_{\mathrm{WD}}})\|$, which, using the approximations above, implies: $\varepsilon \|\nabla F(w_0)\| \gg \lambda \|w^{GF}\|$. Simplifying further (absorbing $\varepsilon$ into a constant), we have the practical guideline: $\lambda \ll \frac{\|\nabla F(w_0)\|}{\|w^{\mathrm{GF}}\|}$. Hence, grokking occurs when the weight decay parameter $\lambda$ is sufficiently small compared to the ratio between the initial gradient magnitude and the norm of the unregularised gradient flow solution.

