# OpenReview forum: "A Theoretical Framework for Grokking: Interpolation followed by Riemannian Norm Minimisation"
_NeurIPS.cc/2025/Conference — NeurIPS 2025 poster_

### Official Review · Reviewer_U5vJ · 2025-06-30

**Clarity:** 4
**Significance:** 3
**Originality:** 3
**Rating:** 5
**Confidence:** 4

**Summary:**

This paper provides a theoretical framework explaining how grokking can arise as a result of weight decay. This framework suggests that gradient flow (GF) on loss $F$ with small weight decay has two phases: in the first phase the trajectory quickly converges to a manifold of stationary points of $F$, and in the second phase the dynamics follows the Riemannian GF on this manifold, minimizing the $\ell_2$-norm of the parameters and slowly converging to an interpolant with a much lower norm, which is believed to generalize better.

**Questions:**

1. In Figure 3, I could not find the loss curves without weight decay. Did they get lost?
2. Could the authors clarify a bit the dependence on initialization scale, to provide some connection with previous work that focuses on NTK vs. rich regimes?

**Ethical Concerns:**

["NO or VERY MINOR ethics concerns only"]

**Final Justification:**

The authors have agreed to fix the small typos and fix the presentation issues indicated by myself and other reviewers. I believe the limitations are well discussed / clear and the scope of the work is generic enough (although it does not encompass everything and does not completely explain grokking). I would treat this as mainly a theoretical contribution, so I would not hold lack of experiments with real-world language models too strongly against this paper. Overall, after reading the other discussions, I keep my score.

**Limitations:**

Yes, the limitations are discussed openly.

**Paper Formatting Concerns:**

No formatting concerns.

**Quality:**

4

**Strengths And Weaknesses:**

## Strengths

In terms of quality:
* The framework is very general optimization (in particular, no structure is assumed on the parameters). The assumptions seem reasonable for a theoretical treatment.
* The treatment is (mathematically) rigorous, the claims are made both informally and formally, and proven in the appendix.
* The theoretical results are supported by simple examples and experiments, covering the standard toy models (quite convincingly).
* Limitations are discussed very openly. No overclaiming noticed.

In terms of clarity, the paper is very well-written. The main theoretical results are split into four propositions and explained well. Toy examples are also clearly reported. There is a nicely written review of related work, with connections well-made (rather than just a list of works).

In terms of significance, this is an important perspective on the reason for grokking, because of its generality.

In terms of originality, this perspective provides significant improvement over the closest works (such as Lyu et al. 2023, Liu et al. 2022c).

## Weaknesses

In terms of significance:
1. This is gradient flow, where weight decay is synonymous with $\ell_2$ regularization. Transformers are usually trained by smoothed versions of signGD, where its role can be different (for example, AdamW converges to KKT points of $\ell_\infty$-norm constrained optimization https://arxiv.org/pdf/2404.04454).
2. As authors point out, grokking has been observed without weight decay, whereas the explanation in this article is completely based on weight decay.
3. Relatedly, classification losses are not handled (also a limitation acknowledged by the authors in the conclusion).

As a minor point, the proof of Proposition 3 in Appendix A.4 got interrupted mid-sentence.

---

> ### Author Rebuttal · Authors · 2025-07-30
>
> We thank the reviewer for their valuable feedback and will carefully address all the remarks in the revised version.
>
> ## Loss curves without weight decay?
>
> For readability, we omitted the loss curves without weight decay from the figure. Overall, the final solution without weight decay closely matches the (regularized) state at time $t_2$​, and the corresponding loss curves align with those shown up to $t_2$​ before remaining essentially flat. We would be happy to include these curves in the revised version if the reviewer believes it would improve the presentation.
>
> ## Clarify dependence on initialization scale
>
> Using the additional page allowed for the camera-ready version, we will include a more detailed discussion on the influence of initialization scale. The initialization scale affects the behavior of the unregularized flow and consequently impacts the first phase of training under regularization. This scale plays a crucial role in the training dynamics, as extensively studied in the implicit bias literature. Although a complete understanding remains elusive, the common consensus—supported by many studies—is that small initialization scales correspond to the rich regime, where implicit bias guides the model toward interpolating solutions with smaller weight norms, which generally leads to better generalization. In contrast, large initialization scales lead to the NTK or lazy regime, where features remain nearly fixed during training. This regime behaves similarly to random feature models and typically results in interpolators with poorer generalization performance.
>
> In our setting, this means that when starting from a small initialization scale, the point $\Phi( w(0) )$ reached after the first phase already has a small norm--potentially corresponding to a KKT point of Equation (7)--and thus no second “grokking” phase occurs (i.e., no significant change happens afterward). Conversely, with a large initialization scale, $\Phi(w(0))$ has a large norm, leading to substantial movement during the second grokking phase.
>
> We will include additional experiments in the appendix of the revised version to illustrate the impact of initialization scale on the grokking phenomenon.

---

> > ### Comment · Reviewer_U5vJ · 2025-08-05
> >
> > Thank you for the reply! I have read the other reviews, and will keep the score as is (recommending acceptance). Please fix the typos/ommissions discussed above.

---

### Official Review · Reviewer_W5fb · 2025-07-02

**Clarity:** 4
**Significance:** 2
**Originality:** 2
**Rating:** 4
**Confidence:** 4

**Summary:**

This paper gives account of grokking as a dynamical phenomenon that occurs over two timescales: when training with very small weight decay $\lambda$, gradient flow first (a) converges to very close to whatever solution it was going to anyways, then (b) slowly drifts towards a local optimum of the $\ell_2$ weight penalty on the manifold of loss minimizers. They prove this for fairly general losses; nothing's specific to deep learning in their analysis.

**Questions:**

I suppose my main questions are conceptual and mostly listed above. The big one (which isn't easy, and which one could and should ask of most papers) is: why does this matter, really? Why does grokking matter, and who does this paper really help? I think a really convincing answer to this actually would get me to increase my score, but that'd be difficult. I'd be curious about the answer regardless of whether or not I'm convinced.

**Ethical Concerns:**

["NO or VERY MINOR ethics concerns only"]

**Final Justification:**

This is a high-quality paper. It solidifies a previously-vague idea. My score isn't higher mostly because I doubt the importance of grokking, and the main idea here is one I'd already heard several times.

**Limitations:**

yep

**Quality:**

4

**Strengths And Weaknesses:**

Some merits of the paper include:
* it's clearly written and easy to follow.
* its conclusion is intuitive (this is emphatically a plus, not a minus), and this makes it easy to skip around and get the gist. the various steps link together in an intuitive way.
* while I didn't check the math, the authors seem to cross their t's and dot their i's; the notation, presentation, etc. seem both mathematically complete and pretty readable, which isn't easy to do.

All considered, to me, this reads as a cogent and well-presented story, and it's nice to have a formal reference for this intuitive idea. Seems to clear a reasonable bar for quality and usefulness, so I recommend acceptance.

...but ya can't have it all, so here are some critiques, which I share with the aim of improving the paper and explaining why my score isn't higher:
* This doesn't seem all that conceptually novel. *That's okay* -- too many papers try way too hard to be conceptually novel and force it -- but I'm just noting it. I've heard this qualitative explanation for grokking before, and the story of this paper was basically what I figured was meant: the weight decay kicks in on a longer timescale, so you get slow movement after naive convergence, and this can slide you over to a better solution. It's nice to see this formalized, especially in deep-learning-independent language, but I'm not sure if there supposed to be any surprises or new insights you get from this analysis.
* I'm not convinced grokking itself is all that interesting or important. I don't count this against the paper for purposes of this review, but it *is* a broader public conversation that should happen somewhere. Why, again, do we think that grokking matters? If anything, the practical utility of this paper seems to be that, having an explanation for grokking, we can stop studying it so intensely, be satisfied that it's just a result of timescale separation, and move on to other things, which we could have done without the formal description of the phenomenon.
* If there were a mystery of grokking, seems like it'd be in the test performance: why, with unregularized GF, we'd converge to a spurious solution, but then converge to the "right" solution with regularization. (My skeptical lean here reflects the fact that this is a much bigger Q and reaches well beyond the phenomenon of grokking.) From this perspective, seems like we've formalized the easy part, but the deeper mystery, except in special cases, is still open.


I hope the authors take the fact that my criticisms are fairly high-level to be praise of the paper: I didn't have to spend time critiquing "implementation details."

---

> ### Author Rebuttal · Authors · 2025-07-30
>
> We thank the reviewer for their valuable feedback and will carefully address all the remarks in the revised version. These questions/remarks are very high level and we share most of these concerns/questionings. We give below our “point of view” on these matters, more than fully supported answers.
>
> ## Why does grokking matter?
>
> In that direction, a recent position paper at ICML 2025 [1] argued that grokking may not be a particularly central phenomenon and suggested that the research community should be cautious about devoting excessive attention to it. However, even within that paper, the authors emphasize that studying such settings can contribute to a broader and deeper understanding of training dynamics in deep learning. We fully agree with this perspective and believe our work advances this broader goal. Our theoretical framework is general enough to capture meaningful and widely applicable behaviors in neural network training.
> While our main results are motivated by the grokking phenomenon, they also offer insights beyond it. Specifically:
> 1. Our analysis provides a general characterization of training dynamics in the small-regularization regime, which can be useful for understanding behavior in a variety of settings—not limited to grokking.
> 2. Although grokking is often discussed in the context of weight decay, our approach can easily be extended to other forms of regularization. In particular, we believe that some empirical observations reported in *“The effect of SAM for deep networks at different stages of training”* [2] for training with Sharpness-Aware Minimization (SAM), may also be interpreted through the lens of grokking, albeit driven by SAM-style regularization rather than standard weight decay.
>
> ## This doesn’t seem conceptually novel
>
> We agree that the intuition that “grokking is due to a first interpolating phase, followed by a second phase where regularization kicks in” has been suggested before, for instance in the OmniGrok paper. Since this idea is natural and makes sense, we were actually surprised not to find any formal mathematical treatment of this intuition in the literature. Our work aims to fill this gap. Moreover, while it is nice to say that “regularization kicks in,” as mathematicians, we appreciate having a more precise description. This is what we provide here, showing that this second phase exactly corresponds to a Riemannian gradient flow for the $\ell_2$​-norm on the interpolation manifold.
>
> Additionally, we felt that this interpretation of grokking was not universally understood, and that the role of weight decay remained broadly unclear, as raised in our “role of weight decay” paragraph.
>
>
> ## we've formalized the easy part, but the deeper mystery, except in special cases, is still open.
>
> Of course, the question of generalization in deep learning is a longstanding and largely unresolved problem that has been extensively studied over many years. In this work, we build upon the commonly held belief that smaller norm weights tend to correlate with better generalization, while acknowledging that this relationship is far from definitive in general settings. Our goal is not to solve this fundamental question, but rather to leverage existing insights on generalization to provide an interpretation of grokking that complements our optimization analysis.
>
> -----------
> # References
>
> [1] Jeffares, A., & van der Schaar, M. *Position: Not All Explanations for Deep Learning Phenomena Are Equally Valuable*. In Forty-second International Conference on Machine Learning Position Paper Track.
>
> [2] Andriushchenko, Maksym, and Nicolas Flammarion. "*Towards understanding sharpness-aware minimization*." International conference on machine learning. PMLR, 2022.

---

> > ### Comment · Reviewer_W5fb · 2025-08-03
> > **rebuttal ack**
> >
> > Thanks for your reply. I agree with most of what you've said, and I'll keep my score as is. As you say, it's nice to formalize folk intuition, and I appreciate that this paper does so in a clear and minimal way.

---

### Official Review · Reviewer_CjUp · 2025-07-02

**Clarity:** 3
**Significance:** 3
**Originality:** 2
**Rating:** 3
**Confidence:** 4

**Summary:**

This paper studies the gradient flow with weight decay in the limit as λ→0. The authors characterize its two-stage behavior: in the first stage it tracks the unregularized gradient flow until it reaches the manifold, and in the second stage it follows the Riemannian flow on that manifold. They use this theoretical framework to explain the phenomenon known as grokking. They verify their findings through synthetic experiments.

**Questions:**

1. How do the authors explain the sudden drop in validation loss observed in grokking, rather than the slow Riemannian flow tracking predicted by their theory?
2. In Section 4.2, the definition of the limit flow appears to follow Li et al. (2021) rather than being original, yet it is not properly cited.
3. Could the authors provide experiments on real-world datasets? Even small-scale ones would be fine.

**Ethical Concerns:**

["NO or VERY MINOR ethics concerns only"]

**Final Justification:**

The author provided a good rebuttal, clearly explaining the plateau period of grokking and providing additional background by connecting it with earlier works. While I still consider the technical novelty and practical significance to be limited, the clarifications have addressed several of my concerns, and based on these considerations, I have raised my rating from 2 to 3.

**Limitations:**

yes

**Quality:**

2

**Strengths And Weaknesses:**

**Strengths:**
- The paper tackles an interesting problem and provides a new theoretical perspective on the grokking phenomenon.
- The writing is fluent, with thorough coverage of related work and background, making it easy to follow.
- The simulation experiments convincingly support the authors’ claims.

**Weaknesses:**
1. The main contribution is not prominent. To my knowledge, similar two-stage gradient flow analyses have been widely explored; although the authors’ focus on the weight-decay variant is novel, their theory relies on the λ→0 limit, which substantially weakens the findings. I question the practical relevance of this assumption. Even theoretically, if λ is a nonzero constant, the gradient flow in the first stage may not converge to the manifold at all. Additionally, the proof of the main result (Proposition 2) heavily relies on the definition and properties of the limit flow from prior work, offering limited technical innovation.
2. The theory does not fully account for grokking. Under the authors’ framework, the second stage should slowly track the Riemannian flow—implying a plateau and gradual change in validation loss—whereas grokking typically shows a long plateau followed by a sudden drop in validation loss.
3. All experiments are synthetic. Experiments on real-world language models are essential.

---

> ### Author Rebuttal · Authors · 2025-07-30
>
> We thank the reviewer for their valuable feedback and will carefully address all the remarks in the revised version.
>
> ## 1. How do you explain the sudden drop in validation loss observed in grokking?
>
> We would like to clarify that the observed drop is not actually sudden, even in practice. It may appear so because, as in most grokking papers, results are plotted in log-time scale. In such plots, the "drop" phase occupies a much smaller portion of the x-axis than the preceding plateau, which can create the impression of a sudden change. However, when viewed on a linear time scale, the drop is more gradual. This is not unique to our experiments but is common across many grokking studies (e.g., Power et al., 2022; Liu et al., 2022c).
>
> Furthermore, Proposition 2 in our paper gives a theoretical explanation for this. It predicts that the second phase of learning occurs within a time interval $[\varepsilon/\lambda, M/\lambda]$, where $\varepsilon$ is small and $M$ is large, both independent of $\lambda$. As a result, on a log scale, the drop appears within an interval of length $\ln⁡(M/\varepsilon)$, while the plateau spans $[\Theta(1),\Theta(1)/\lambda]$, corresponding to a length of $\ln(1/\lambda)$ on the same scale. This explains why the drop looks sudden relative to the plateau in log-scale plots. However, in linear time, the drop spans a length of a similar order of magnitude than the preceding plateau, giving a different visual impression.
>
> ## 2. Relation to Li et al (2021)
>
> We acknowledge that Li et al. (2021) revived the “two-stage” gradient flow analysis, which is the foundation of our proof of Proposition 2. We do not claim novelty of this analytical framework. In fact, this approach dates back to Katzenberger (1990) and was further developed in more modern contexts by Fatkullin et al. (2010). The limit flow $\Phi$ is a standard object in continuous-time optimization and already appears in Katzenberger’s work. We will clarify these connections more explicitly in the revision. In particular, we will emphasize how Li et al. (2021) recently highlighted the power of this technique.
>
> At a high level, our contribution differs from Li et al. (2021) in that we explain a slow drift induced by a **deterministic regularization**, whereas their analysis focuses on stochastic effects. Additionally, we relate this drift directly to the **grokking phenomenon**, which to our knowledge has not been previously addressed in this context. From a technical standpoint, our setting is simpler and allows for a more detailed analysis:
> 1. Rather than invoking Katzenberger (1990) as a black-box result, we provide a self-contained and accessible proof that builds on Falconer (1983), whose ideas also underlie the works of Katzenberger and Li et al.
> 2. Moreover, while prior analyses typically assume initialization close to the interpolation manifold, our approach handles **general initialization**. We rigorously analyze the first phase, where the flow approaches the manifold, and then establish the transition to the second phase (drift along the manifold). Dealing with this transition is non-trivial and, as we mention following Proposition 2, is the main technical novelty in this proof.
>
> ## 3. Experiments on real-world datasets.
>
> Many empirical studies have already demonstrated the grokking phenomenon on real-world datasets. We reference several of these in the paragraph “Grokking in experimental works” in Section 2, and we do not believe that reproducing similar experiments would yield additional insights.
>
> Our objective is not to contribute further real-world evidence, but rather to validate the coherence between our theoretical results and controlled toy simulations. These simplified settings allow us to precisely monitor the relevant quantities and illustrate our theoretical findings.
>
> Naturally, we would be happy to include any additional references of interest if you have suggestions.
>
> ## 4. The theory relies on the $\lambda \to 0$ limit.
>
> Our work indeed focuses on the $\lambda \to 0$ limit, as it allows for a simple and tractable analysis. Providing clean results with a constant $\lambda > 0$ would require stronger assumptions and likely lead to much more complex proofs.
>
> Regarding the statement that
>
> > even theoretically, if  $\lambda$ is a nonzero constant, the gradient flow in the first stage may not converge to the manifold at all,
>
> although this is true, we can still show that the flow reaches a point that is *close to the manifold*, and upper bound the distance as a function of $\lambda$.
>
> In particular, Equation (9) in the Appendix (proof of Proposition 1) provides an upper bound on the deviation between the regularized and unregularized flows. This allows us to bound the distance of the unregularized flow to the manifold for finite $\lambda$ and a carefully chosen time $t$ (as detailed in the proof of Lemma 2). While we acknowledge that this bound may not be tight enough for practical use, it does offer some theoretical control in the finite $\lambda$ setting.
>
> Also note that in all our synthetic experiments, which are run for non-zero but small values of $\lambda$, the training loss goes down to *nearly* zero, indicating that the iterates *nearly converge* towards the manifold.
>
> Besides, although our results are asymptotic, they appear to match what we observe empirically for fixed, small $\lambda$.
>
> We hope our response clarifies the points raised and appreciate the reviewer’s consideration of these explanations.

---

> > ### Comment · Reviewer_CjUp · 2025-08-05
> >
> > Thank you for your detailed and thoughtful responses. As most of my concerns have been addressed, I will raise my score.

---

### Official Review · Reviewer_Bgxe · 2025-07-03

**Clarity:** 3
**Significance:** 3
**Originality:** 4
**Rating:** 4
**Confidence:** 4

**Summary:**

The paper lays down purely theoretical justification for grokking phenomenon, a two-phase learning behavior where generalization occurs long after training loss reaches zero. With assumptions on stability, smoothness, and non-pathological behavior of the underlying models, the authors has built a purely theoretical justification of two-phase learning behavior under small l2 weight decay. The paper also supplies small-scale experiments to demonstrate the theoretically discussed behaviors.

**Questions:**

These are the questions that I do not think to be appropriate to be classified as the weaknesses but do still believe to be important. Below are request for clarifications of my possible misunderstandings.

1. I may have lost but is t≈1/λ transition repeatedly appearing throughout the manuscript justified? Is this transition boundary sharp or gradual in practice?
2. How small the weight decay should be in order to be applied by this theory? Do the authors have a clue when does the grokking start to not happen as we increase the weight decay OR decrease the initial weights?

The next question is from my pure curiosity and therefore the answer will not affecting the overall scores.

1. In the Line 347-349 and the referred Appendices C and D.2, the authors have argued that grokking may promote sparse estimators. From my point of view, grokking seems to “correct” the bad initialization at the beginning of the training. As demonstrated in OmniGrok, sufficiently small weight initializations can quickly set the model to the so-called “rich region”, and **de-grok** the model. This implies that we have to be careful about the features that seem to be **unlocked** by the grokking phenomenon. Is this the “grokking” phenomenon that “unlocks” certain features of the model that would not be possible without “large weight initialization & grokking”, or is it just the problem-intrinsic feature that can also be achieved by **de-grokked** models?

**Ethical Concerns:**

["NO or VERY MINOR ethics concerns only"]

**Final Justification:**

I have read through all the rebuttal and other reviewer's comments. The authors have provided a rich set of explanations that has resolved many of my initial concerns. However, I believe that without further closing the gap between the Transformer architecture and the suggested theory in an explicit way (e.g., establishing Assumption 2 as the authors has suggested), the practical value of this work remains not significant enough. Therefore, I will keep my score.

**Limitations:**

Yes.

**Paper Formatting Concerns:**

None.

**Quality:**

3

**Strengths And Weaknesses:**

## Strengths

1. The paper presents (almost) model-agnostic explanation of grokking, achieving generalizability and bridges many of the empirically found grokking examples on diverse set of architectures and data (e.g., OmniGrok).
2. The assumptions underlying the theory is reasonable, and embraces rich set of models.
3. The paper is well-formatted and pleasurable to read.

## Weaknesses

Majority of my concerns are inquiries rather than criticisms. But here are some points the authors may consider in order to improve the presentation and connection to the practical branch of works.

1. It would be nicer to have a list of real-world functions/models/losses/training scenarios that the underlying assumptions (Assumption 1, 2) hold. It is roughly argued for generality of the assumption throughout page 4. I believe, however, listing concrete examples, e.g., MLP with Lipschitz-continuous activations like smooth rectifying or sigmoidal nonlinearities, sigmoid after quadratics like self-attention layer, etc., will greatly enhance the impact of the theoretical findings, and help the follow-up works to find out where should they start working on.
2. I believe it will be much better if there is another list that those assumptions do not hold. For example, Line 129-131 and line 328-330 were graceful, but a dedicated list of exclusion maybe in the separate limitation section would be more elegant.
3. The first report on grokking phenomenon was on Transformer network, which contains not only residual MLP but also residual multi-head attention. So it will be nicer to have this application explained by the proposed theory, too.

Overall, I believe the paper is already in a good shape regarding the theoretical novelty. However, I have several concerns related to presentation and demonstration. I will raise my score if those issues are resolved. Please also refer to the questions section.

---

> ### Author Rebuttal · Authors · 2025-07-30
>
> We thank the reviewer for their valuable feedback and will carefully address all the remarks in the revised version.
>
> ## When do Assumptions 1, 2 hold?
>
> In short: our assumptions are quite generic. They hold for most basic neural network architectures with smooth activations, up to some important caveats (in general, they might only hold *locally*, and after excluding certain degenerate configurations).
>
> **Assumption 1** is pretty mild: it holds for all architectures that use differentiable activations (i.e. that do not use ReLU). Although ReLU is a very common activation, many architectures now use differentiable variants of ReLU such as GeLU and thus satisfy Assumption 1. The ‘restrictive’ part of Assumption 1 is that it assumes that $w_{GF}$ is bounded: this does not hold when training with cross entropy loss (or other exponential tail classification losses) on separable datasets. Apart from this case  (i.e. in regression tasks or when the convergence point does not interpolate the data), Assumption 1 holds.
>
> **Assumption 2** (Morse-Bott property) is the more restrictive one. It requires the Hessian matrix $\nabla^2 F(w)$ to have a constant rank on $\mathcal{M}$. In general, this is not true globally, as $\mathcal{M}$ can have degenerate points. However, in most cases it holds *locally* around *almost every point* of $\mathcal{M}$. For example:
> - for the "diagonal neural network" toy model, it holds if we restrict to points with nonzero coordinates,
> - for matrix completion, it does if we exclude matrices with degenerate rank,
> - it also holds *locally almost everywhere* for wide MLPs, ConvNets and ResNets with smooth activations; see for instance Liu, Zhu and Belkin (2021)
>
> In our work, we preferred to assume that the Morse-Bott property holds globally to derive simpler results and analysis. Similar global assumptions were considered by the works of Li et al (2021), Shalova et al (2024), Fatkullin et al (2010). Although it is not exactly true in practice, we believe it still allows to derive meaningful analyses.
> Note that we could extend our results to the local case, provided that the iterates remain in the non-degenerate region.
>
> We will add more discussion about these assumptions in the revised version, using the extra page allowed for the camera ready version.
>
> ## Grokking with attention explained by the proposed theory
>
> We agree that applying our theoretical framework to analyze grokking in Transformers is a promising and worthwhile direction.
>
> As mentioned above, our assumptions are quite generic. They can be shown to hold for attention-based architectures, although establishing Assumption 2 rigorously requires extra work.
>
> However, to show that our results lead to grokking in this case, we need to understand the generalization properties of minimal-norm solutions for Transformers.
> In Section 5, we focus on simple neural network architectures, for which there already exists a rich body of work characterizing implicit bias and minimal-norm interpolators.
> Extending this analysis to attention-based models is a highly non-trivial task that calls for a study of its own.
>
> ## Is the transition at $t=\frac{1}{\lambda}$ theoretically justified? Is the transition sharp or gradual?
>
> Yes, our work gives a precise statement showing that the transitions happens at $t=\frac{1}{\lambda}$.
> This follows from the change of timescale for $\tilde{w}$, as defined on line 208: we will make this clearer in the revised version.
>
> Indeed, Proposition 2 implies that at time $\varepsilon/\lambda$, the regularized flow remains close to the limit of the unregularized flow, which interpolates but does not generalize well in the high-norm initialization regime. In contrast, at time $\Theta(1/\lambda)$, the flow approaches the limit of the Riemannian gradient flow on the interpolation manifold. This point has smaller norm and, as a result, is expected to generalize better.
>
> As explained in our answer to Reviewer CjUp, this transition is gradual (both theoretically and empirically), but appears sharp when time is plotted in log-scale. Though mathematically incorrect (in the sense that the transition is not “immediate” even in log scale), people tend to say it looks sharp. We claim this is mostly because relatively to the long plateau before the transition, the transition indeed looks “rapid”, i.e. “sharp”.
>
> ## How small should the weight decay be to observe grokking?
>
> Indeed, our results are derived in the asymptotic regime $\lambda \to 0$, since this allows for a tractable analysis. Extending the theory to a fixed $\lambda > 0$ is considerably harder. That said, we can offer some heuristic analysis regarding how small $\lambda$ needs to be for grokking to emerge.
>
> Grokking depends on a clear separation between two phases: an initial phase where the training loss rapidly converges to zero, and a second, slower phase driven by weight decay, during which test performance improves. For grokking to be observable, the gradient flow should approach a near-stationary point before weight decay begins to significantly influence the dynamics.
>
> To formalize this intuition, we can define two characteristic times: $t_{GF}$​, the convergence time of the unregularized gradient flow, measured as the time at which the gradient norm substantially decreases relative to its initial value; and a second timescale of order $1/\lambda$, associated with the onset of regularization effects. When $\lambda$ is small enough that $t_{GF}\ll 1/\lambda$, we expect to observe grokking-like behavior. Specifically, for some threshold $\varepsilon \ll 1$ (e.g., $\varepsilon=0.01$): $ \Vert \nabla F(w_{t_{GF}})\Vert  \approx \varepsilon \Vert \nabla F(w_0)\Vert $. Now let $t_{WD}$ denote the time when weight decay kicks in: i.e. when the magnitude of the unregularised gradient becomes comparable to the magnitude of the weight decay term: $\Vert \nabla F(w_{t_{WD}})\Vert  \approx \lambda \Vert w_{t_{WD}}\Vert $. Since at time $t_{WD}$, the solution is close to the gradient flow solution $w^{GF}$, we can consider $\Vert w_{t_{WD}}\Vert  \approx \Vert w^{GF}\Vert $. The condition for grokking to occur (i.e., a plateau in test loss followed by improvement) is thus that $t_{GF} \ll t_{WD}$. Translating this condition in terms of gradients, we obtain:
> $\Vert \nabla F(w_{t_{GF}})\Vert \gg \Vert \nabla F(w_{t_{WD}})\Vert $, which, using the approximations above, implies:
> $  \varepsilon \Vert \nabla F(w_0)\Vert \gg \lambda \Vert w^{GF}\Vert $. Simplifying further (absorbing $\varepsilon$ into a constant), we have the practical guideline: $\lambda \ll \frac{\Vert \nabla F(w_0)\Vert }{\Vert w^{GF}\Vert }$. Hence, grokking occurs when weight decay is sufficiently small relative to ratio of the initial gradient magnitude over the norm of the unregularised gradient flow solution.
>
> **Empirically**, we observed the grokking effect in our experiments for values of $\lambda$ up to $10^{-2}$. For such relatively large values, the initial plateau becomes noticeably shorter. As $\lambda$ increases further, the two phases begin to slightly overlap, making the separation between them less distinct (but still present).
>
> ## Question on feature unlocking and de-grokking
>
> > Is this the “grokking” phenomenon that “unlocks” certain features of the model that would not be possible without “large weight initialization & grokking”, or is it just the problem-intrinsic feature that can also be achieved by de-grokked models?
>
> Our interpretation of the grokking phenomenon is as follows: when initialized with large weights and trained without weight decay, the model does not necessarily learn meaningful features in the sense of prioritizing the relevant ones. Instead, its output depends on a broad set of features, including many that are not informative. In contrast, when regularization is introduced (or in the so-called "rich" regime), we do not believe that the model *“unlocks”* entirely new features that were absent in the earlier regime. Rather, it learns to concentrate weight primarily on the relevant features–those that contribute meaningfully to generalization–while down-weighting or discarding the irrelevant ones.
> This contrasts with the unregularized setting (with large initialization), where the model tends to distribute weight more uniformly across both relevant and irrelevant features, with the latter possibly dominating due to their greater frequency.
>
> In this context, the notion of "sparsity" refers to the model ultimately placing significant weight on only a small subset of useful features. Prior to grokking, by contrast, the model assigns non-negligible weights to a broader set of features, many of which are not essential for generalization.

---

> > ### Comment · Reviewer_Bgxe · 2025-08-07
> >
> > Thank you for the elaborative and intuitive responses. Most of my initial concerns are resolved. However, I still believe that even though it is nontrivial, founding theoretical ground for the Transformer-type architectures is important for a highly valued academic work in this field. So I am afraid I cannot raise my score than my initial recommendation, though I recommend this to be accepted.

---

### Decision · Program_Chairs · 2025-09-17

**Decision:**

Accept (poster)

**Comment:**

This paper gives a general, optimization-centric account of grokking by proving that gradient flow with small L2 weight decay exhibits two phases: a fast approach to an interpolation manifold, followed by a slow Riemannian gradient flow that minimizes the L2 norm along that manifold. Commendations mainly include broad, mostly model-agnostic theoretical framework with reasonable assumptions (Bgxe, U5vJ), mathematically careful and rigorous (W5fb, U5vJ), empirical support that matches the theory (CjUp, U5vJ, Bgxe), and useful perspective/advance relative to prior explanations of grokking (CjUp, U5vJ). Main concerns include practical relevance of the lambda -> 0 regime and limited novelty beyond prior two-stage analyses (CjUp, W5fb), gap to real-world practice: experiments limited to synthetic data; unclear coverage of transformers/attention (CjUp, Bgxe), and explaining the observed "sudden" drop vs predicted gradual drift (CjUp, Bgxe). Overall, I think this is a nice paper deserving to be accepted.